# Bile salt hydrolase catalyses formation of amine-conjugated bile acids

Bipin Rimal[1,19], Stephanie L. Collins[2,19], Ceylan E. Tanes[3], Edson R. Rocha[4], Megan A. Granda[1], Sumeet Solanki[5], Nushrat J. Hoque[6], Emily C. Gentry[7,8,18], Imhoi Koo[9], Erin R. Reilly[2], Fuhua Hao[1], Devendra Paudel[10], Vishal Singh[10], Tingting Yan[11], Min Soo Kim[2], Kyle Bittinger[3], Joseph P. Zackular[12,13], Kristopher W. Krausz[11], Dhimant Desai[14], Shantu Amin[14], James P. Coleman[4], Yatrik M. Shah[5], Jordan E. Bisanz[2,15], Frank J. Gonzalez[11], John P. Vanden Heuvel[1,16], Gary D. Wu[17], Babette S. Zemel[3], Pieter C. Dorrestein[7,8], Emily E. Weinert[2,6] & Andrew D. Patterson[1,2,15]✉

Bacteria in the gastrointestinal tract produce amino acid bile acid amidates that can affect host-mediated metabolic processes[1–6]; however, the bacterial gene(s) responsible for their production remain unknown. Herein, we report that bile salt hydrolase (BSH) possesses dual functions in bile acid metabolism. Specifically, we identified a previously unknown role for BSH as an amine *N*-acyltransferase that conjugates amines to bile acids, thus forming bacterial bile acid amidates (BBAAs). To characterize this amine *N*-acyltransferase BSH activity, we used pharmacological inhibition of BSH, heterologous expression of *bsh* and mutants in *Escherichia coli* and *bsh* knockout and complementation in *Bacteroides fragilis* to demonstrate that BSH generates BBAAs. We further show in a human infant cohort that BBAA production is positively correlated with the colonization of *bsh*-expressing bacteria. Lastly, we report that in cell culture models, BBAAs activate host ligand-activated transcription factors including the pregnane X receptor and the aryl hydrocarbon receptor. These findings enhance our understanding of how gut bacteria, through the promiscuous actions of BSH, have a significant role in regulating the bile acid metabolic network.

Bile acids are host–microbiota cometabolites that facilitate dietary lipid absorption and regulate lipid, glucose and xenobiotic metabolism[7–9]. In humans and rodents, the primary bile acids cholic acid (CA) and chenodeoxycholic acid (CDCA) are synthesized in the liver from cholesterol. Most of the CA and CDCA is then conjugated with glycine or taurine in a two-step reaction catalysed by bile acid–coenzyme A (CoA) synthetase and bile acid–CoA:amino acid *N*-acyltransferase[10] (Fig. 1a). Conjugated primary bile acids, predominantly glycocholic acid (GCA) and taurocholic acid (TCA), are stored in the gallbladder until the consumption of a meal triggers their secretion into the duodenum. Bile salt hydrolase (BSH), an enzyme commonly found in bacteria inhabiting the small intestine and colon, hydrolyses the amide linkage of conjugated bile acids[11] (Fig. 1b). Posthydrolysis, bacteria further metabolize the bile acid backbone to generate secondary bile acids such as deoxycholic acid (DCA) and lithocholic acid, which are either excreted in the faeces or reabsorbed, conjugated and recycled by the host[5,12]. BSH activity has been associated with colonization resistance to *Clostridium difficile*[13], weight gain[14] and regulation of circadian rhythm[15]. Furthermore, the addition of a recombinant BSH cocktail changes the pathogenesis of *C. difficile*[16], which was previously attributed to BSH deconjugation activity.

Recent studies demonstrated that bacteria, in addition to the host, have bile acid amine *N*-acyltransferase activity, with dozens of bacterially conjugated amino acid amidates identified[1–4,6,17]. The enzymes responsible for these conjugation reactions are unknown. Furthermore, bacterial bile acid amidates (BBAAs) are abundant in the gastrointestinal tract[18] and have been associated with human inflammatory bowel disease[3]. Although bacterial reconjugation of bile acids shapes the bile acid pool, how this process affects host–microbiota communication is not understood. Some BBAAs activate host bile acid nuclear receptors like the farnesoid X receptor (FXR) and pregnane X receptor (PXR)[1,3], but an assessment of other receptors has not been completed. We demonstrate herein that bacterial BSHs re-amidate bile acids to produce BBAAs in addition to their known role in hydrolysing conjugated bile

[1]Department of Veterinary and Biomedical Sciences, Pennsylvania State University, University Park, PA, USA. [2]Department of Biochemistry and Molecular Biology, Pennsylvania State University, University Park, PA, USA. [3]Division of Gastroenterology, Hepatology, and Nutrition, Children's Hospital of Philadelphia, Philadelphia, PA, USA. [4]Department of Microbiology and Immunology, Brody School of Medicine, East Carolina University, Greenville, NC, USA. [5]Department of Molecular & Integrative Physiology and Internal Medicine, Division of Gastroenterology, University of Michigan, Ann Arbor, MI, USA. [6]Department of Chemistry, Pennsylvania State University, University Park, PA, USA. [7]Skaggs School of Pharmacy and Pharmaceutical Sciences, University of California San Diego, San Diego, CA, USA. [8]Collaborative Mass Spectrometry Innovation Center, Skaggs School of Pharmacy and Pharmaceutical Sciences, University of California San Diego, San Diego, CA, USA. [9]Huck Institutes of the Life Sciences, Pennsylvania State University, University Park, PA, USA. [10]Department of Nutritional Sciences, Pennsylvania State University, University Park, PA, USA. [11]Laboratory of Metabolism, Center for Cancer Research, National Cancer Institute, National Institutes of Health, Bethesda, MD, USA. [12]Division of Protective Immunity, Children's Hospital of Philadelphia, Philadelphia, PA, USA. [13]Department of Pathology and Laboratory Medicine, Perelman School of Medicine, University of Pennsylvania, Philadelphia, PA, USA. [14]Department of Pharmacology, Penn State University College of Medicine, Hershey, PA, USA. [15]One Health Microbiome Center, Huck Life Sciences Institute, University Park, PA, USA. [16]INDIGO Biosciences, Inc., State College, PA, USA. [17]Division of Gastroenterology and Hepatology, Perelman School of Medicine, University of Pennsylvania, Philadelphia, PA, USA. [18]Present address: Department of Chemistry, Virginia Tech, Blacksburg, VA, USA. [19]These authors contributed equally: Bipin Rimal, Stephanie L. Collins. ✉e-mail: adp117@psu.edu

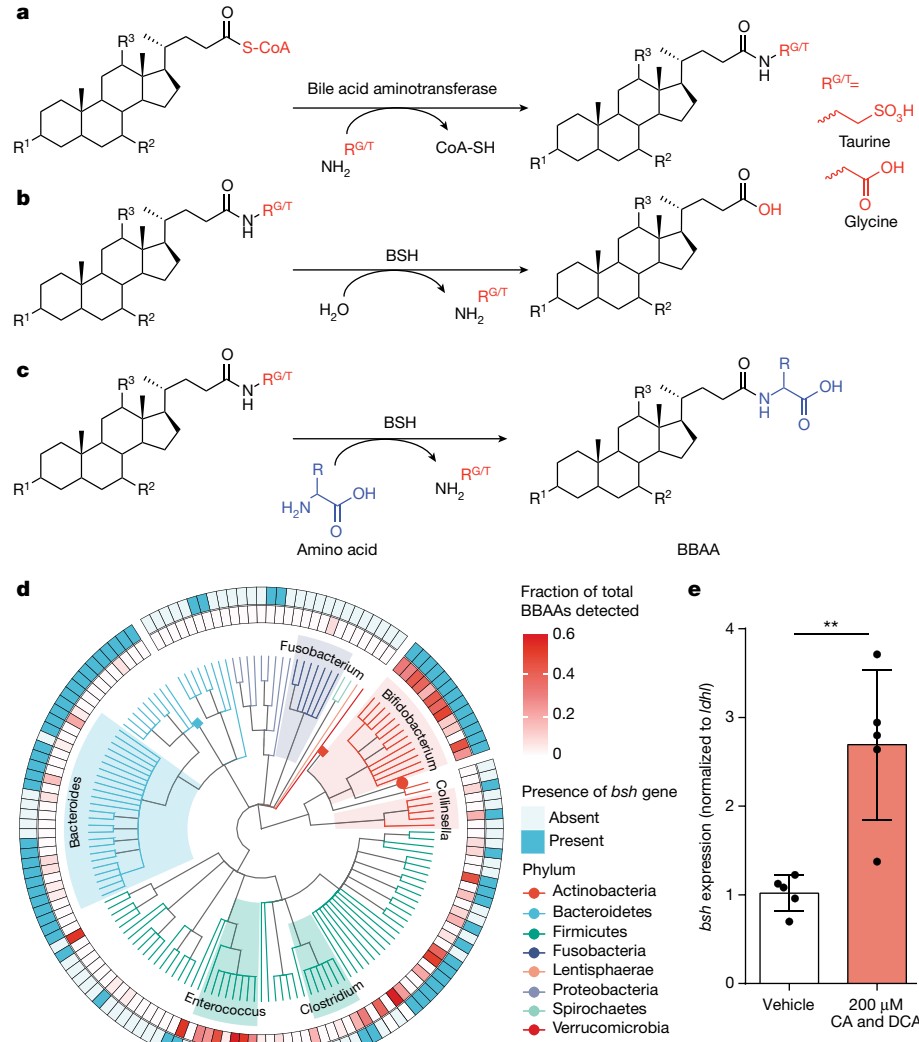

**Fig. 1 | Amidation and de-amidation reactions of bile acids are associated with bacterial *bsh*. a**, Bile acid–CoA:amino acid *N*-acyltransferase activity with the substrate depicted as a general bile acid structure modified with CoA. **b**, Deconjugation reaction by bacterial BSH of host-derived GCA or TCA. **c**, The BSH amino acid *N*-acyltransferase activity characterized in this study, with the conjugated bile acids GCA and TCA as examples. This activity leads to the biosynthesis of BBAAs. **a–c**, R[1]–R[3] represent potential sites of backbone hydroxylation; R[G/T] represents glycine and taurine. **d**, Correlation between presence or absence of the *bsh* gene with the ability of various bacterial taxa to produce BBAAs. **e**, RT-qPCR of *B. longum* NCTC 11818 supplemented with 200 µM each of CA and DCA. Expression of *bsh* was compared to vehicle using the $2^{-\Delta\Delta Ct}$ method normalized to the reference gene *ldhl*, and significance was determined using a two-tailed *t*-test ($P = 0.0026$**). Bar plots designate the mean ± s.d., $n = 5$.

acids (Fig. 1c). We found that BBAAs are produced in early life during expansion of the BSH-expressing gut microbiota. In addition, we report that in cell culture models, unique collections of BBAAs activate host ligand-activated receptors to potentially mediate host–microbiota communication.

## BSH correlates with BBAA production

Diverse bacteria from the human gastrointestinal tract produce BBAAs[2,3]. We therefore set out to discover the gene(s) responsible for the production of BBAAs. It was recently demonstrated that gut bacteria can amidate CA with phenylalanine, tyrosine and leucine[1]. Additionally, in a screen of 202 human-associated bacterial isolates, production of BBAAs from a mixture of CA and DCA was reported[3]. Another study screened 72 bacterial isolates and found that 25 strains were able to conjugate DCA, CDCA or CA to produce at least one BBAA[2]. We compared the genomes of the isolates involved in these two studies with their production of BBAAs to identify possible candidate gene(s)

responsible for BBAA production. We found that the bile salt hydrolase (*bsh*) gene was significantly associated ($P = 9.4 \times 10^{-6}$, phylogenetic linear regression) with BBAA production (Fig. 1d and Extended Data Fig. 1). The association is exemplified in the genera *Bifidobacterium* and *Enterococcus*, which are efficient producers of BBAAs and also widely harbour the *bsh* gene, whereas *Fusobacterium* spp., which lack the *bsh* gene, do not make BBAAs. Overall, with few exceptions, bacterial isolates that produce at least one BBAA have a *bsh* homologue.

To further explore the involvement of *bsh* and other genes in bile acid conjugation, we performed RNA sequencing on the BBAA-producing strain *Bifidobacterium longum*, National Collection of Type Cultures (NCTC) 11818, treated with 100 µM unconjugated bile acids (CA and DCA, equimolar). Global transcriptional profiling identified 1,929 genes, of which 89 were differentially expressed (adjusted $P \leq 0.05$ and $\log_2 FC \leq -1.5$ or $\geq 1.5$), including *bsh*, which was significantly upregulated ($\log_2 FC = 2.14$, adjusted $P = 6.52 \times 10^{-85}$) (Extended Data Fig. 2 and Supplementary Table 1). We also validated the RNA sequencing results with quantitative reverse transcription polymerase chain reaction

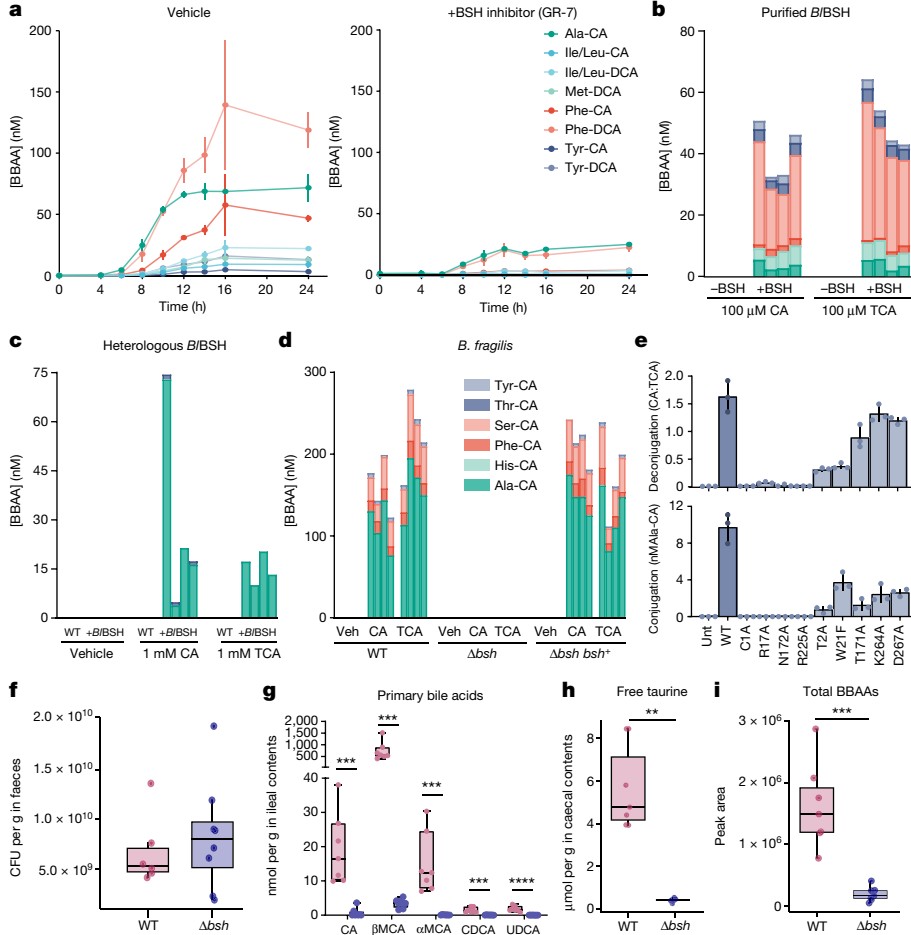

**Fig. 2 | Bacterial BSH has bile acid amine N-acyltransferase activity. a**, The pan-BSH inhibitor GR-7 attenuates conjugated bile acid production in *B. longum* NCTC 11818 cultures. Mean BBAA concentrations, quantified by targeted liquid chromatography with tandem mass spectrometry (LC–MS/MS), are shown with vertical lines indicating s.d. (*n* = 3 biologically independent bacterial cultures per treatment group). **b**, Purified *Bl*BSH enzyme synthesizes BBAAs from 100 μM CA or TCA in vitro. **c**, *E. coli* transformed with a *Bl*BSH expression vector produces BBAAs from 1 mM CA or TCA. **d**, BSH knockout in *B. fragilis* ablates the aminotransferase activity observed in wild-type (WT) *B. fragilis* NCTC 9343 with 1 mM CA or 1 mM TCA supplementation. Bars in **b**–**d** represent individual biological replicates (*n* = 4), with heights indicating BBAA concentrations. **e**, *E. coli* BL21(DE3) expressing mutated *Bl*BSH at key active site residues exhibited altered deconjugation and conjugation activities. Deconjugation, indicated by the CA:TCA ratio, was assessed in M9 media with 1 mM TCA. Conjugation, measured through Ala-CA production, was evaluated in Luria–Bertani media with 1 mM TCA using LC–MS/MS. Controls included untransformed *E. coli* (Unt) and *E. coli* expressing unmutated *Bl*BSH (WT). Bars represent mean values ± s.d. (*n* = 3 biologically independent bacterial cultures). **f**–**i**, Bile acid profile of germ-free C57BL/6 J mice monocolonized with *B. fragilis* WT or *Δbsh* strains (*n* = 7 for WT, *n* = 8 for *Δbsh* groups). CFUs per gram of *B. fragilis* in faeces after 7 days (**f**). Ileal contents primary bile acids quantified by LC–MS/MS (CA $P$ = 2.3 × 10$^{-4}$***, βMCA $P$ = 1.3 × 10$^{-4}$***, αMCA $P$ = 3.7 × 10$^{-4}$***, CDCA $P$ = 1.1 × 10$^{-4}$, UDCA $P$ = 2.7 × 10$^{-5}$****) (**g**). Free taurine levels in caecal contents quantified using $^1$H nuclear magnetic resonance (NMR) ($P$ = 4.27 × 10$^{-4}$***) (**h**). Sum of peak areas for all BBAAs in ileal contents measured by LC–MS/MS ($P$ = 3.1 × 10$^{-4}$***) (**i**). **f**–**i**, box plots depict first quartile and third quartile with median as centre and min–max values for whiskers, except for outliers calculated as data points ±1.5× interquartile range. Significant differences ($P$ < 0.001***, $P$ < 0.0001****) were determined by two-tailed *t*-tests. MCA, muricholic acid; UDCA, ursodeoxycholic acid.

(RT-qPCR), confirming that *bsh* messenger RNA was increased 2.7-fold ($P$ = 0.0026) (Fig. 1e). Previously, *bsh* expression was reported to be upregulated in response to a mixture of conjugated and unconjugated bile acids, but the response was attributed solely to the presence of conjugated bile acids[19,20]. Therefore, we sought next to examine the possibility that BSH has amine *N*-acyltransferase activity with unconjugated bile acids as the substrate.

## BSH is a bacterial *N*-acyltransferase

We hypothesized that pharmacologic inhibition of the BSH enzyme would eliminate the production of BBAAs from unconjugated bile acids. The pan-BSH inhibitor gut restricted-7 (GR-7) (ref. 21) was used to inhibit BSH of B. *longum* NCTC 11818 treated with equimolar 100 μM CA and DCA. We monitored the production of BBAAs from 0 to 24 h

with and without GR-7. Addition of 100 μM GR-7 resulted in a significant decrease in the production of BBAAs, particularly phenylalanine- and alanine-conjugates, as compared to vehicle-treated controls (Fig. 2a). The attenuated BBAA synthesis was not due to bacterial growth inhibition because colony-forming units (CFUs) were equal with or without GR-7 (Extended Data Fig. 3).

Given that *Bifidobacterium* strains are difficult to manipulate genetically[22], we investigated the necessity of BSH using *Bacteroides fragilis* NCTC 9343, which has been extensively studied for its role in secondary bile acid metabolism and is more compatible with genetic manipulation. We generated a complete *bsh* deletion strain (*Δbsh*) and another strain that complemented the *bsh* knockout in trans (*Δbsh bsh⁺*). We then treated the bacterial strains with 1 mM CA or 1 mM TCA for 16 h to assess whether differences could be observed in BBAA production by BSH from unconjugated or conjugated bile acid substrates. Deletion of

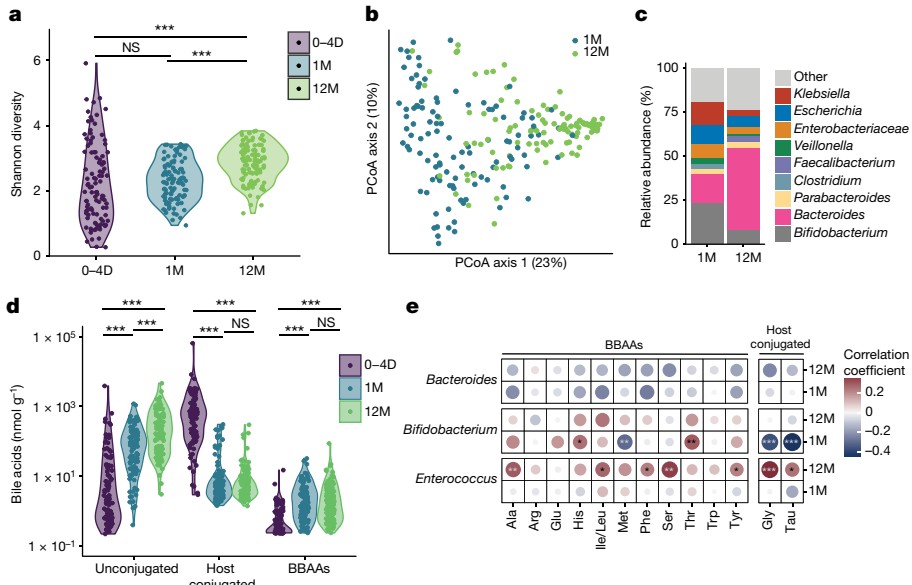

**Fig. 3 | BBAA production increases with infant microbiota development.**
**a**, Microbiota diversity measured by Shannon diversity index of infant faecal samples acquired at 0–4 days (0–4D), one month (1M) and 12 months (12M) after birth, $n = 108$. Significant differences between time points were determined by linear mixed effects model (0–4D versus 12M $P = 4.4 \times 10^{-7}$***, 1M versus 12M $P = 2 \times 10^{-5}$***). **b**, Principal coordinate analysis plot of the Bray–Curtis dissimilarity of infants 1M and 12M old. Significant differences were determined by permutational multivariate analysis of variance (PERMANOVA) test on Bray–Curtis distances ($P = 0.001$***). **c**, Barplot depicting faecal microbiota composition of 1M and 12M infants, including all genera greater than 5% mean relative abundance in either time point. **d**, Faecal concentrations of total unconjugated, host glycine and taurine-conjugated and BBAAs from 0–4D to 12M, quantified using LC–MS/MS. Significant differences between time points were determined by linear mixed effects models, with log(bile acids) as outcomes, time as predictor and subject information as random effect ($P < 0.001$***). Several comparisons were adjusted using the Benjamini–Hochberg method. **e**, Associations of BBAAs and glycine-/taurine-conjugated BAs with relative *Enterococcus*, *Bifidobacterium* and *Bacteroides* abundances using Spearman correlation ($P < 0.05$*, $P < 0.01$**, $P < 0.001$***). NS, not significant.

*bsh* eliminated the production of BBAAs from both CA and TCA, whereas complementation rescued BBAA production (Fig. 2d). These observations support our hypothesis that BSH has amine *N*-acyltransferase activity.

To test whether BSH is sufficient for the production of BBAAs, we cloned the *bsh* gene from *B. longum* NCTC 11818 into the pET-28b(+) vector (*Bl*BSH) and overexpressed in *Escherichia coli* BL21(DE3). Heterologous expression of the enzyme in *E. coli*, which lacks endogenous *bsh*, confers amine *N*-acyltransferase activity. Incubation of the *E. coli-Bl*BSH with 1 mM CA or 1 mM TCA produced BBAAs, whereas the untransformed *E. coli* did not produce BBAAs (Fig. 2c).

Next, we purified *Bl*BSH and assayed for BBAA production from 100 μM CA or TCA and a 250 μM equimolar mixture of amino acids. Both CA and TCA treatment yielded BBAAs, but not all amino acids were conjugated, demonstrating substrate selectivity by BSH. The large proportion of alanine and serine conjugates produced by *B. fragilis* (Fig. 2d), *Bl*BSH-expressing *E. coli* (Fig. 2c) and purified *Bl*BSH (Fig. 2b) indicate that the enzyme prefers amino acids with smaller side chains. However, the greater diversity of BBAAs produced by purified *Bl*BSH in vitro (Fig. 2b) compared to *Bl*BSH-expressing *E. coli* (Fig. 2c) is probably a result of the different amino acid profiles in *E. coli* (alanine is present at about a 100-fold greater concentration than serine)[23].

Although we observed BSH amine *N*-acyltransferase activity with both CA and TCA, the conjugation mechanism is unclear. On the basis of the known mechanistic details of BSH's hydrolytic activities and it belonging to the N-terminal nucleophile superfamily of enzymes, the terminal cysteine Cys[1] of *Bl*BSH is predicted to be the primary catalytic residue, although other active site residues (for example, Thr[2], Trp[21], Thr[171], Asn[172] and Arg[225]) may also be important for BSH activity or substrate specificity[24]. We performed site-directed mutagenesis on nine residues proximal to the active site of *Bl*BSH to identify which are necessary for *N*-acyltransferase activity in BSH (Fig. 2e). The *N*-acyltransferase

activity of BSH closely mirrored the deconjugation activity with each mutation, with C1A, R17A, N172A and R225A all being essential for its function. None of the mutations inhibited expression of the *Bl*BSH protein (Extended Data Fig. 4).

To support the relevance of bacterial BSH in producing BBAAs in vivo, we colonized germ-free mice with the wild-type or Δ*bsh* strain of *B. fragilis* described above. Both strains equally colonized the mouse gastrointestinal tract seven days after gavage (Fig. 2f), and there was no change in body weight between the groups (Extended Data Fig. 5). The bile acid pool of Δ*bsh*-colonized mice reflected a perturbation in bile acid hydrolysis, as the release of unconjugated primary bile acids was reduced and the production of BBAAs attenuated (Fig. 2g,i). Similarly, free taurine was also significantly higher in caecal contents of wild-type-colonized mice compared to Δ*bsh*-colonized mice (Fig. 2h). This observation is consistent with hydrolysis of taurine-conjugated bile acids and the formation of BBAAs. Importantly, we demonstrate that colonization with wild-type *B. fragilis* is key to the production of BBAAs, as BBAA levels are significantly depleted in Δ*bsh*-colonized mice (Fig. 2i). Therefore, in the mouse gastrointestinal tract, colonization with BSH-positive bacteria is required for bile acid hydrolysis and re-amidation.

## BBAA levels coincide with microbiota development

During the first year of life, the gut microbiota undergoes rapid expansion and maturation. To understand the biological relevance of BBAAs, we quantified bile acids including BBAAs from the stool of infants followed from birth to 12 months[25]. Metagenomic sequencing showed an increase in microbial diversity, measured by the Shannon Index, from birth through 12 months (Fig. 3a). Variation in sample diversity between 0 and 4 days is probably due to the rapid maturation of the gut microbiota. Principal coordinates analysis of Bray–Curtis dissimilarity supports

the differences in beta-diversity between one-month- and 12-month-old infants (Fig. 3b, permutational multivariate analysis of variance (PERMANOVA) $P = 0.001$). The *Bacteroides* population expanded from one to 12 months, whereas the relative abundance of *Bifidobacterium* was reduced (Fig. 3c). As the microbiota evolved with age, we observed a shift from host-conjugated to unconjugated primary bile acids due to the colonization of *bsh*-positive bacteria (Fig. 3d). The levels of BBAAs also increased from 0–4 days to 12 months after birth (Fig. 3d). Further stratifying the samples collected at 0–4 days, the levels of several BBAAs (Ala-CA, Glu-CDCA, Tyr-CA, His-CA, Ile/leu-CA, Trp-CA, Ser-CA and Tyr-CDCA) were positively correlated with the hours elapsed after birth (Extended Data Fig. 6). We correlated the abundance of *Bacteroides*, *Bifidobacterium* and *Enterococcus* with levels of host-conjugated (Gly and Tau) bile acids and BBAAs. *Bacteroides* abundance was not correlated with BBAA production. In 12-month-old infants, *Enterococcus* abundance was positively correlated with host-conjugated bile acids and BBAAs including Ala, Ile/Leu, Phe, Ser and Tyr (Fig. 3e). *Bifidobacterium* abundance was negatively correlated with host-conjugated bile acids in one-month-old infants and positively correlated with bile acid conjugates of His and Thr (Fig. 3e). These data support our findings that BSHs from *Bifidobacterium* species are amine *N*-acyltransferases that produce BBAAs in humans. To our knowledge, this is the first demonstration connecting BBAA production with BSH-expressing bacteria during human infant development.

## BBAAs activate host transcription factors

To demonstrate potential biological relevance of BSH-mediated BBAA production, we assessed in cell culture models the activation of VDR, CAR3, PXR and AHR in a cell-based reporter assay (Extended Data Fig. 7). Most of the BBAA activators of the human bile acid receptor FXR were conjugates of CDCA (5 of 8, $P < 0.0001$), probably reflecting the potency of CDCA as an FXR agonist[26]. Bile acids conjugated with glutamate activated human FXR, CAR3, PXR, AHR and PPARα ($P < 0.0001$). Therefore, we selected Glu-CA and Glu-CDCA for further exploration in organoid models. We treated organoids cultured from mouse and human small intestinal crypts with 50 μM or 100 μM Glu-CA or Glu-CDCA and measured the expression of FXR, CAR, PXR and AHR target genes. In mouse organoids, Glu-CA and Glu-CDCA upregulated *Cyp3a11*, *Cyp1a1* and *Bsep* expression, indicating that PXR and AHR were transcriptionally activated (Extended Data Figs. 8a and 9a). Responses in human organoids were less pronounced, but *CYP1A1* expression was also upregulated with 100 μM Glu-CDCA treatment ($P < 0.01$, Extended Data Figs. 8b and 9b). We previously showed that Glu-conjugated bile acids were activators of human PXR in vitro[14], and herein, we observed activation of AHR by both bacterially conjugated and conventional bile acids in vitro. Further studies, both in vitro and in vivo, are required to demonstrate whether BBAAs affect the metabolism of substrates, drugs and carcinogens regulated by PXR[26] and AHR[27].

## Conclusion

We identified a previously undescribed role for BSH as an amine *N*-acyltransferase that produces BBAAs in mice and humans. These BBAAs may facilitate communication between the microbiota and host through the activation of the human ligand-activated transcription factors. As there are hundreds of different BSHs found in microbiota that synthesize different BBAAs[3], BSHs may be instrumental in fine tuning different pathways that promote health or disease. Historically, BSH knockout or inhibition has been associated with modulation of inflammation, cancer and liver disease without knowledge of BSH amine *N*-acyltransferase activity. Future studies must re-evaluate the contribution of BBAA production on disease outcomes related to BSH activity.

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

## Methods

### Correlation of BSH and ability to produce BBAAs

To check for the association of BSH and BBAA production, we leveraged liquid chromatography with tandem mass spectrometry (LC–MS/MS) data from two published datasets involving 202 Human Microbiome Project (HMP) bacterial isolates in ref. 3 and 69 representative gut bacteria members in ref. 2. We combined the two datasets. For comparison between bacteria from the studies, we normalized the dataset on the basis of the proportion of BBAAs detected from the total number of BBAAs tested in each study. Next, we downloaded the assembled genomes of the bacteria from the National Center for Biotechnology Information (NCBI) genome database. A phylogenetic tree was obtained from the NCBI taxonomy database using taxids in the study and visualized using the ggtree[28] package in R. We annotated the genomes using Prokka[29] and checked for the presence or absence of the *bsh* gene. Briefly, Prokka uses several databases (Uniprot, RefSeq and Pfam) in a hierarchical manner to annotate coding sequences in the genome. Genomes with coding sequences annotated as choloylglycine hydrolase (*cbh*) or having its domain were deemed as having the *bsh* gene or its homologue, whereas genomes lacking these annotations were deemed lacking the gene. To test for correlation, we used the phylolm[30] package in R to fit a following phylogenetic linear regression model:

$$BBAAs \approx bsh + 1,$$

where **BBAAs** is the vector of proportions of the BBAAs detected for bacteria in each study and **bsh** is the vector of presence or absence of the *bsh* gene.

### RNA sequencing sample preparation and analysis

*B. longum* subsp. *longum* NCTC 11818 was acquired from ATCC, and 30 µl of an overnight culture of *B. longum* was inoculated into 3 ml of brain–heart infusion (BHI) broth (BD Difco) supplemented with 0.05% w/v cysteine, 5 µg ml$^{-1}$ hemin and 1 µg ml$^{-1}$ vitamin K (Sigma). Equivalent volumes of 1 M NaOH (control) or CA and DCA in 1 M NaOH at a final concentration of 200 µM were added. Cultures were grown to mid-exponential phase (approximately 6 h) at 37 °C in an anaerobic chamber filled with 20% $CO_2$ and 5% $H_2$ in $N_2$ gas. Then 4 ml of RNAprotect (Qiagen) was added to 2 ml culture and incubated for 5 min at room temperature. Samples were centrifuged at 5,000*g* for 5 min at 4 °C, and the bacterial pellets were flash-frozen in liquid nitrogen and stored at −80 °C. Pellets were resuspended in 1 ml TRIzol Reagent and transferred to Lysing Matrix E tubes for homogenization at 2 × 20 s cycles at 6,500*g*. Next, 200 µl chloroform was added, vortexed for 15 s and incubated at room temperature for 10 min. Samples were centrifuged for 15 min at 16,000*g*, 4 °C, and 500 µl upper aqueous phase was transferred to a fresh tube containing 500 µl of 100% ethanol. Samples were then transferred to a Purelink spin column, centrifuging for 15 s at 12,000*g* to elute flowthrough. Columns were washed with 350 µl Wash Buffer I and removed by centrifugation for 15 s at 12,000*g*. Contaminating DNA was degraded by adding 80 µl of deoxyribonuclease (DNase) solution (10 µl DNase in 1× reaction buffer) and incubating at room temperature for 15 min. A second wash with Wash Buffer I was then performed. Columns were washed twice with 500 µl Wash Buffer II, eluting each time by centrifugation for 15 s at 12,000*g*. Purified RNA was eluted with 30 µl RNase-free water by centrifugation for 1 min at 12,000*g* into a fresh tube.

RNA samples were submitted to the Microbial Genome Sequencing Center for sequencing. Samples were DNase treated with Invitrogen DNase (RNAse-free). We prepared the library using Illumina's Stranded Total RNA Prep Ligation with Ribo-Zero Plus Kit and 10 bp IDT for Illumina indices. Sequencing was done on a NextSeq2000 to yield 2 × 50 bp reads. Demultiplexing, quality control and adaptor trimming were performed with bcl-convert (v.3.9.3).

For the analysis, the complete reference genome and annotation of *B. longum* NCTC 11818 were obtained from the NCBI Assembly database. Obtained demultiplexed reads were checked for quality and filtered using fastp (v.0.12.4) (ref. 31). Filtered reads were aligned with the reference genome using bowtie2 (v.2.2.5) (ref. 32). Obtained alignments were sorted using samtools (v.1.10) (ref. 33). Qualimap (v.2.2.2) (ref. 34) was used to check for the quality of alignments. featureCounts (v.2.0.1) (ref. 35) was used to generate counts of reads uniquely mapped to annotated genes. These raw counts were then normalized and analysed for differential gene expression using the DESeq2 package (v.1.35.0) (ref. 36) in RStudio.

qScript cDNA SuperMix (Quantabio) was used for the reverse transcription of 250 ng RNA template in a 20 µl reaction for RT-qPCR. Primer sequences were designed using PrimerBlast for *B. longum bsh* and lactate dehydrogenase (*ldhl*) (Supplementary Table 2). We performed the RT-qPCR with 1 µl complementary DNA (cDNA) template and 0.4 µM of each primer on a QuantStudio 3 using fast cycling parameters with PowerUp SYBR detection reagent.

### *B. longum* BSH inhibition

*B. longum* subsp. *longum* NCTC 11818 was acquired from ATCC, and 30 µl of an overnight culture of *B. longum* was inoculated into 3 ml of BHI broth (BD Difco) supplemented with 0.05% w/v cysteine, 5 µg ml$^{-1}$ hemin and 1 µg ml$^{-1}$ vitamin K (Sigma). Next, 2 µl of the overnight culture was added to 196 µl fresh supplemented-BHI media, and 1 µl of 1 M NaOH (vehicle) or 20 mM CA and DCA in 1 M NaOH at a final concentration of 100 µM were added to the plate. To inhibit BSH, 1 µl of 20 mM GR-7 in dimethyl sulfoxide (DMSO) was added to the treatment wells to a final concentration of 100 µM. Equivalent volume of DMSO was added to the control wells. To monitor the production of BBAAs, samples were taken out at 4, 6, 8, 10, 14, 16 and 24 h after incubation and flash-frozen in liquid nitrogen until extraction for LC–MS/MS quantification of bile acids.

### Heterologous expression and purification of *B. longum* BSH in *E. coli*

The gene for *B. longum* BSH (NC_015067.1, codon-optimized) cloned into pET-28b(+) using NdeI and XhoI restriction sites was synthesized by GenScript. The resulting plasmid was transformed into *E. coli* Tuner (DE3) cells (Novagen) by means of electroporation at 2,500 V, and positive transformants were selected on Luria–Bertani media containing 50 µg ml$^{-1}$ kanamycin. Expression strains were grown in yeast extract media (45 g yeast extract (Research Products Int.), 1.6 g $KH_2PO_4$ (Research Products Int.), 13.0 g $K_2HPO_4$ (Research Products Int.) and 1% glycerol (v/v) (Sigma-Aldrich) per 1 l) at 37 °C to late exponential phase (optical density at 600 nm ($OD_{600}$) = 0.8). The cells were cooled to 18 °C before induction with 0.1 mM isopropyl β-d-1-thiogalactopyranoside (Research Products Int.) and allowed to express BSH for 18–20 h before being collected by centrifugation (4,000*g* at 4 °C, 20 min). The resulting cell pellets were frozen at −80 °C until use.

The cells were resuspended in buffer A (50 mM Tris (Research Products Int.), 300 mM NaCl (Research Products Int.), 20 mM imidazole (Sigma-Aldrich), pH 7.4) with protease inhibitors benzamidine HCl (Research Products Int.) and Pefabloc SC (Sigma-Aldrich) and disrupted by means of sonication (QSonica Sonicator Q500) for 6 min on ice (30 s pulse, 30 s pause, at 60% amplitude). The resulting lysate was centrifuged at 130,000*g* in a Beckman Optima L-90X ultracentrifuge at 4 °C for 1 h.

The following purification steps were performed at 4 °C. Supernatant was applied to a HisPur Ni-column (Fisher Scientific) pre-equilibrated with buffer A. The column was washed with 20 column volumes of buffer, and then the BSH protein was eluted with 5 column volumes of buffer B (50 mM Tris, 300 mM NaCl, 250 mM imidazole, pH 7.4). The eluent was dialysed into the final storage buffer (50 mM Tris, 50 mM NaCl, 5% glycerol (v/v), pH 7.0) overnight at 4 °C. The resulting protein

solutions were concentrated using Amicon Ultra 15 ml centrifugal filters (Millipore) with a molecular weight cutoff of 10 kDa, flash-frozen in liquid nitrogen and stored at −80 °C.

## In vitro purified enzyme assay
Purified *Bl*BSH enzyme was diluted to 10 mg ml$^{-1}$ in phosphate buffer (pH 7.0). An equimolar mixture of 21 amino acids was prepared in H$_2$O (5 mM each of L-alanine, L-arginine, L-asparagine, L-aspartic acid, L-cysteine, L-glutamine, L-glutamic acid, glycine, L-histidine, L-isoleucine, L-leucine, L-lysine, L-methionine, L-phenylalanine, L-proline, L-serine, L-threonine, L-tryptophan, L-tyrosine, L-taurine and L-valine). Each reaction mixture contained 178 µl phosphate buffer (pH 6.0), 10 µl of 5 mM amino acids mixture, 2 µl of 100 mM bile acid (CA or TCA in DMSO) or vehicle (DMSO) and 10 µl of the 10 mg ml$^{-1}$ enzyme. The reaction mixture was incubated for 90 min at 37 °C. The reaction was stopped by adding 800 µl ice-cold 100% methanol containing 0.5 µM deuterated bile acid internal standards.

## Construction of *B. fragilis* NCTC 9343 *Δbsh* mutant
A 1,797 bp DNA fragment containing the N-terminal region of the BSH gene was PCR amplified from the *B. fragilis* NCTC 9343 using primers BSH-NT-FOR and BSH-NT-REV. The N-terminal fragment was cloned into the SphI/BamHI sites of the pFD516 vector. Then a 1,734 bp DNA fragment containing the C-terminal region was PCR amplified with primers BSH-CT-FOR and BSH-CT-REV2 and cloned into the SacI site of pFD516. The new construct, pER363, contains a deletion of 765 bp DNA fragment from the *bsh* gene. A 3,533 bp BglII/SalI DNA fragment from pER363 containing the *Δbsh* construct was cloned into the BamHI/SalI sites of the suicide vector for allelic replacement in *Bacteroides*, pLGB13 to construct pER-366. pER-366 was mobilized from *E. coli* S17-1 λpir into *B. fragilis* NCTC 9343 by biparental mating as described previously. Transconjugants were selected on brain–heart infusion supplemented (BHIS) plus 0.5% yeast extract plates containing 200 µg ml$^{-1}$ gentamycin and 10 µg ml$^{-1}$ erythromycin. Four transconjugant colonies were grown overnight on BHIS broth with 200 µg ml$^{-1}$ gentamycin (erythromycin was omitted). Then 10 µl of an overnight culture was spread on BHIS with 0.5% yeast extract plates containing 200 µg ml$^{-1}$ gentamycin and 100 ng ml$^{-1}$ anhydrotetracycline. Transconjugant colonies were streaked on fresh BHIS with 0.5% yeast extract plates containing 200 µg ml$^{-1}$ gentamycin and 100 ng ml$^{-1}$ anhydrotetracycline. PCR amplification using BSH-mutcheck-FOR and BSH-mutcheck-REV primers was used to identify chromosomal double-crossed over genetic recombination. The *B. fragilis* NCTC 9343 *Δbsh* mutant isolates (BER-279) were also tested for loss of erythromycin resistance and respiratory test.

## Genetic complementation of *Δbsh* mutant
pER-300 containing *bsh* operon in the pNBU2-*bla-ermGb* vector was mobilized into BER-279 by biparental mating. Transconjugants were selected on BHIS plus 0.5% yeast extract plates containing 200 µg ml$^{-1}$ gentamycin and 10 µg ml$^{-1}$ erythromycin (BER-280).

## Site-directed mutagenesis of *Bl*BSH active site amino acid residues
The pET-28b(+)-*Bl*BSH vector constructed above was mutated using the Q5 Site-Directed Mutagenesis Kit according to the manufacturer's instructions (New England Biolabs). The NEBaseChanger tool was used to design primers (Supplementary Table 2) and calculate the appropriate annealing temperatures for PCR. Following PCR mutagenesis and treatment with kinase, ligase and DpnI, the products were transformed into NEB 5-alpha-competent *E. coli* by heat shock at 42 °C, and positive transformants were selected on Luria–Bertani media containing 50 µg ml$^{-1}$ kanamycin. Individual colonies were selected, and successful mutations were confirmed by QIAprep Spin Miniprep kit (Qiagen) preparation of plasmid DNA followed by

Sanger sequencing from the T7 promoter region. Positively identified plasmid DNA was stored at −20 °C. To express the proteins, mutant pET-28B(+)-*Bl*BSH plasmids were transformed into *E. coli* BL21(DE3) cells by electroporation at 2,500 V, and positive transformants were selected on Luria–Bertani media containing 50 µg ml$^{-1}$ kanamycin. To measure TCA deconjugation, transformed *E. coli* were cultured in M9 media supplemented with 1 mM TCA. To measure CA conjugation, transformants were cultured in Luria–Bertani media supplemented with 1 mM CA. All transformants were grown at 37 °C and 250 rpm until OD$_{600}$ ≈ 0.6 was reached; then protein was induced by the addition of 0.1 M isopropyl β-d-1-thiogalactopyranoside and grown for a further 4 h. *E. coli* were extracted for LC–MS/MS according to the protocol below.

## Extraction and LC–MS/MS quantification of bile acids from bacterial cultures
For bacterial cultures, 200 µl of the culture following growth was lysed by three freeze–thaw cycles in liquid nitrogen. Next, 150 µl of each lysate was transferred to a 1 ml capacity 96-well plate, and 600 µl of ice-cold 100% high-performance liquid chromatography (HPLC)-grade methanol containing 0.5 µM deuterated bile acid internal standards was added and mixed thoroughly by pipetting. The plate was then incubated on ice for 20 min. Samples were sonicated for 10 min in a sonicating water bath and then centrifuged for 15 min at 2,000 rpm at 4 °C. Then 200 µl of each supernatant was transferred to a fresh 1.5 ml microcentrifuge tube and dried using a speedvac. Dried extracts were resuspended in 200 µl HPLC-grade methanol and sonicated for 10 min before transfer to autosampler vials for LC–MS/MS.

Quantitative LC–MS/MS analyses were performed using a Waters ACQUITY Ultra-performance liquid chromatography (UPLC) system coupled with a Waters Xevo TQ-S Triple Quadrupole mass spectrometer (Waters). Chromatographic separation was achieved using an ACQUITY BEH C8 (2.1 × 100 mm, 1.7 µm) UPLC column (Waters) heated to 60 °C. Samples (1 µl) were injected into the column and eluted with 90% mobile phase A (1 mM ammonium acetate in 9% acetonitrile, pH 4.15) and 10% mobile phase B (1:1 acetonitrile:isopropanol) for 0.1 min. Mobile phase B was increased in consecutive linear gradients from 10% to 35% over 9.15 min, 35% to 85% B over 2.25 min and 85% to 100% B over 0.3 min. Then 100% B was held for 0.6 min followed by a linear gradient to 10% B over 0.1 min and held for 2.5 min, for a total of 15 min. The flow rate was modified throughout the run starting with 300 µl per min for 9.25 min and then increasing to 325 µl per min over 2.25 min and 500 µl per min over 0.6 min. The flow rate was held at 500 µl per min for 0.3 min and then reduced to 300 µl per min over 0.4 min and held for 2.2 min. MS analyses were carried out using electrospray ionization in positive ion mode using the following settings: capillary voltage 3.5 kV, cone voltage 20 V, source temperature 150 °C, desolvation temperature 300 °C, desolvation gas flow 540 l h$^{-1}$ and cone gas flow 150 l h$^{-1}$. BBAAs were identified by multiple reaction monitoring transitions from parent [M + H]$^+$ ions to daughter [M + H]$^+$ ions corresponding to the amino acid fragment (Supplementary Table 3). Peak identification and processing was done using MassLynx software (Waters). Peak areas for each conjugated bile acid were normalized to peak areas from deuterated internal standards of GCA-d4 or GDCA-d4, matching the bile acid. Quantification was performed against a standard curve of authentic standards at eight concentrations ranging from 7.81 nM to 1 µM.

## Monocolonization of GF mice
Fifteen male 8- to 10-week-old C57BL/6 J mice were maintained under germ-free conditions in positive-pressure isolators (Class Biologically Clean). Gnotobiotic experiments were conducted under the Pennsylvania State University Institutional Animal Care and Use Committee approved protocol 202101826. Mice were singly housed with ad libitum access to water and food. Sterility of mice was ensured by qPCR and

aerobic/anaerobic culture with rich media. Mice were administered $10^8$ CFUs of either *B. fragilis* NCTC 9343 or the *Δbsh* mutant, with mice monitored daily. No adverse effects on mouse weight or behaviour were observed. After 7 days of colonization, mice were euthanized and ileal and caecal tissue and contents collected. To quantify *B. fragilis* colonization, faecal samples were resuspended in phosphate-buffered saline (PBS) and serially diluted before quantification on BHI plates using the drop-plate method.

### Extraction and untargeted LC–MS/MS profiling of monocolonized mice

Bile acids were extracted from mouse ileal contents according to ref. 37. Briefly, 25 mg of ileal contents were weighed in 1.5 ml screw-cap homogenization tubes, and approximately 50 μl of 0.1 mm zirconia beads were added. Next, 1 ml of ice-cold methanol containing 0.5 μM deuterated internal bile acid standards was added. Samples were then thoroughly homogenized in Precellys 24 tissue homogenizer at 20 s × 2 at 6,500 rpm. Homogenized samples were then freeze–thawed three times in liquid nitrogen and centrifuged at 15,000 rpm at 4 °C. Then 200 μl of the supernatant was transferred to autosampler vials for LC–MS/MS analysis.

We performed LC–MS/MS analyses using a Thermo Fisher Scientific Orbitrap Exploris 120 Mass Spectrometer system coupled with a Vanquish ultra-high-performance liquid chromatography system (Thermo Fisher Scientific). Chromatographic separation was carried out using an ACQUITY BEH C18 UPLC column (2.1 × 100 mm, 1.7 μm; Waters). The column was maintained at a temperature of 60 °C. Mobile phase solvent A was a mixture of 9% acetonitrile and 1 mM ammonium acetate with a pH of 4.15, and solvent B was a 1:1 ratio of acetonitrile and isopropanol. Samples (5 μl) were injected into the column, and the initial mobile phase was 10% B, kept for 0.1 min. Mobile phase B increased in consecutive linear gradients to 35% at 9.25 min and to 85% B at 11.5 min and was held at 85% for 3.4 min. It then decreased to 10% B at 15.1 min and was held at 10% B for 2.9 min. The flow rate of the mobile phase was 0.3 ml min$^{-1}$. Heated electrospray ionization was used with the following settings: spray voltages of 3.5 kV for positive charge and 2.5 kV for negative charge, sheath gas at 50 Arb, auxiliary gas at 10 Arb, sweep gas at 1 Arb, ion transfer tube temperature 325 °C and vaporize temperature 350 °C. MS settings included an Orbitrap resolution of 120,000, a scan range ($m/z$) from 150 to 1,200 and an RF lens of 60%. MS2 spectra were data-dependently acquired using higher-energy collisional dissociation with normalized collision energies of 20%, 35% and 50% and a resolution of 60,000. Data were analysed using MS-DIAL (v.4.9.221218).

### Amino acid quantification of caecal content by $^1$H NMR

Roughly 50 mg of caecal content was weighed in a 2.0 ml screw-cap microcentrifuge tube and mixed with 1.2 ml of 0.1 M phosphate buffer ($K_2HPO_4$:$NaH_2PO_4$ = 4:1, pH = 7.4) containing 50% $D_2O$ and 0.005% (w/v) of sodium 3-(trimethylsilyl) propionate-2,2,3,3-d4 as internal standard. Samples were homogenized and then subjected to two freeze–thaw cycles in liquid nitrogen. After centrifuging at 17,000$g$ for 10 min at 4 °C, 550 μl of clear supernatant was transferred to 5 mm NMR tubes for NMR analysis.

The one-dimensional $^1$H spectra of caecal content extracts were recorded at 298 K by using a Bruker Avance NEO 600 MHz NMR spectrometer (Bruker Biospin) equipped with a 5 mm TCI cryoprobe with enhanced sensitivity for $^1$H. The noesygppr1d pulse sequence was used, and spectral acquisition parameters were as follows: 64 scans were collected into 64,000 data points using 90° pulses, four dummy scans, 20 ppm spectral width. All spectra were analysed with Chenomx NMR Suite (v.9.02). After automatic processing, the phase, baseline and internal standard were checked and modified manually for each spectrum for quality assurance. The amino acids were identified and quantified using a built-in metabolite library.

### Cell-based reporter assays

Reporter assay kits for human farnesoid X receptor (hFXR, cat. no. IB00601); vitamin D receptor (hVDR, IB00701); constitutive androstane receptor variant 3 (hCAR3, IB00901); pregnane X receptor (hPXR, IB07001); aryl hydrocarbon receptor (hAhR, IB06001); peroxisome proliferator-activated receptors alpha (hPPARα, IB0011), beta/delta (hPPARβ/δ, IB0012) and gamma (hPPARγ, IB0010); and mouse FXR (mFXR, M0060) and PPAR alpha (mPPARα, M0011), beta/delta (mPPARβ/δ, M0012) and gamma (mPPARγ, MR0010) were purchased from INDIGO Biosciences, Inc. Agonism assays were performed with test compounds at a single concentration (50 μM) with a concurrent dose-response of reference ligands following the manufacturer's instructions. Briefly, each reporter assay cell suspension was thawed by the addition of cell recovery medium (CRM), and 100 μl of the cell suspension were used per well. Then 100 μl of compound screening medium supplemented with test compounds were added and incubated at 37 °C, 5% $CO_2$ for 24 h. The next day, the medium was removed, detection reagents were added and luminescence was measured (Synergy H4 Hybrid Multi-Mode Microplate Reader). Data were expressed relative to vehicle-treated cells and $\log_{10}$-transformed for visualization. Differences between treatments were determined using analysis of variance (ANOVA) followed by Dunnett's post hoc test (JMP Pro 15, SAS Institute). Significant differences were determined when $P < 0.05$. Non-linear regression and EC50 calculations were performed with Prism v.6.0 (GraphPad Software, Inc.).

### Enteroid model

Normal (de-identified) human small intestinal samples were established and cultured in high-Wnt-containing medium (human colonoid medium) as previously described[38]. Enteroids were treated with 50 μM and 100 μM Glu-CA or Glu-CDCA for 18 h, the medium was removed, and enteroids were washed twice and collected in PBS. Matrigel and PBS were carefully removed, and enteroids were lysed and extracted using a PureLink RNA mini kit according to the manufacturer's instructions. qScript cDNA SuperMix (Quantabio) was used for the reverse transcription of 500 ng RNA template in a 40 μl reaction for RT-qPCR. We performed the RT-qPCR with 1 μl cDNA template and 0.4 μM of each primer (Supplementary Table 2) on a QuantStudio 3 using fast cycling parameters with PowerUp SYBR detection reagent.

### IGRAM cohort

African American women in their third trimester of pregnancy were enroled as part of a prospective, longitudinal, observational maternal–infant cohort study (Infant Growth and Microbiome (IGRAM))[25]. The study was approved by the Institutional Review Board at Children's Hospital of Philadelphia. Stool samples from infants (56 male and 52 females) were collected at birth, 1 month and 12 months of life.

### LC–MS/MS quantification of bile acids in IGRAM stool samples

A separate method was used for the quantification of bile acids in the IGRAM cohort. Bile acids were extracted from infant stool according to ref. 37. Briefly, 25 mg of stool was weighed in 1.5 ml screw-cap homogenization tubes, and approximately 50 μl of 0.1 mm zirconia beads were added. Next, 1 ml of ice-cold methanol containing 0.5 μM deuterated internal bile acid standards was added. Samples were then thoroughly homogenized in Precellys 24 tissue homogenizer at 20 s × 2 at 6,500 rpm. Homogenized samples were then freeze–thawed three times in liquid nitrogen and centrifuged at 15,000 rpm at 4 °C. Then 200 μl of the supernatant was transferred to autosampler vials for LC–MS/MS analysis in a Thermo Fisher Scientific TSQ Quantis Plus mass spectrometer.

UPLC parameters were modified from the method of ref. 39. LC–MS/MS analyses were performed using a Vanquish ultra-high-performance liquid chromatography system coupled with a TSQ Quantis Plus mass

spectrometer (Thermo Fisher Scientific). Chromatographic separation was achieved using a BEH C18 (2.1 × 100 mm, 1.7 μm) UPLC column (Waters) heated to 60 °C. Then 2 μl of samples was injected into the column with a flow rate of 0.4 ml min$^{-1}$. Mobile phase A was 1 mM ammonium acetate and 0.1% formic acid in water, and mobile phase B was 1 mM ammonium acetate and 0.1% formic acid in 225:25:3 methanol:acetonitrile:water. The chromatographic gradient for a total 22 min method was 65% A at 0 min, 65% A at 1 min, 40% A at 2 min, 35% A at 9 min, 0% A at 15 min, 0% A at 20 min and 65% A at 20.5 min until 22 min. MS analyses were carried out using a triple quadrupole mass spectrometer with electrospray ionization in negative ion mode using the following settings: spray voltage 3.5 kV, sheath gas-flow rate 45 arb, aux gas-flow rate 10 arb, sweep gas-flow rate 1 arb, ion transfer tube temperature 325 °C and vaporizer temperature 350 °C. BBAAs were identified by selected reaction monitoring (SRM) mode with a chromatographic peak width of 12 s, Q1 and Q3 resolution of 1.2 full-width at half-maximum and collision gas pressure of 1.5 mTorr. SRM transitions from parent [M-H]$^-$ ions to daughter [M-H]$^-$ ions corresponded to the loss of bile acids (Supplementary Table 4). Peak areas for each conjugated bile acid were normalized to peak areas from deuterated internal standards of GCA-d4, GUDCA-d4 or GDCA-d4, matching the bile acid. Quantification was performed against a standard curve of authentic standards at eight concentrations ranging from 7.81 nM to 1 μM.

### Shotgun metagenomics of IGRAM stool samples

DNA was extracted from faecal samples and from negative controls using the PowerSoil-htp kit (MO BIO Laboratories) following the manufacturer's instructions, with the optional heating step included (note that MO BIO has since been purchased by QIAGEN; the extraction kit is now sold as the DNeasy PowerSoil HTP 96 Kit). Shotgun libraries were generated from 1 ng of DNA using the NexteraXT kit (Illumina). Libraries were sequenced on the Illumina HiSeq using 2 × 125 bp chemistry in High Output mode. In each plate, two positive controls containing a mix of *Vibrio campbellii* and PhiX lambda phage were added.

Shotgun metagenomic data were analysed using Sunbeam v.2.0.1 (ref. 40). Low-quality sequences were trimmed using Trimmomatic with default parameters[41]. Sequences with low complexity and sequences that align to the human and phiX genomes were removed from further analysis. The abundance of bacteria was estimated using Kraken[42]. Sample similarity was assessed by Bray–Curtis dissimilarity and visualized on a principal coordinate plot. Difference in community composition was assessed using the PERMANOVA test. The Shannon diversity metric was used to calculate the alpha diversity. Differences in bile acid levels across the time points were calculated using linear mixed effects models. Log-transformed bile acid levels were used as the outcome, the time points were modelled as the predictor, and the subject information was added as the random effect.

### Reporting summary

Further information on research design is available in the Nature Portfolio Reporting Summary linked to this article.

### Data availability

RNA sequencing datasets are deposited in NCBI SRA Database under bioproject PRJNA878764. BBAA quantification data were obtained from earlier published studies (https://doi.org/10.21203/rs.3.rs-820302/v1 and https://doi.org/10.1128/mSystems.00805-21), and corresponding bacterial genomes for comparative genomics were obtained from NCBI RefSeq. Peak areas from bile acid quantification are found in the Supplementary Information. For the human study, raw triple quadrupole MS data have been deposited in the Mass Spectrometry Interactive Virtual Environment (MassIVE) under accession no. MSV000093144. Structures of *B. longum* and *Clostridium perfringens* BSH were acquired from the Protein Database (2HF0 and 2BJF, respectively). The IGRAM

study enrolled pregnant African American mothers and their newborn infants. The purpose of the research study was to learn more about the bacteria normally living in the child's gut, how it is transferred from mother to child and whether it affects the child's growth in the first three years of life. The IGRAM data used in this publication were consistent with the stated purpose of the research. The Institutional Review Board-approved consent documents included language that allowed participants to indicate whether they would like to have their information included in future research. Subjects may participate in the original research without their information (even if de-identified) being included in future research. Therefore, the data submitted to the repository (SRA accessions PRJNA1042647 and PRJNA557731) were limited to those individuals who consented to future use of their data and are not the entire dataset used in the analyses presented here. To request the complete dataset, contact B. Zemel (zemel@chop.edu) or K. Bittinger (bittingerk@chop.edu) with a summary of how the data will be used and how it is consistent with the goals of the study. The request will be reviewed by the study team at Children's Hospital of Philadelphia. If approved, the data will be made available within 1 month at no cost. Source data are provided with this paper.

### Code availability

The code used for analysis and generating the figures are found at GitHub (https://github.com/PattersonLab-PSU/BBAAs-paper) and Zenodo (https://doi.org/10.5281/zenodo.10072966).

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

**Acknowledgements** We thank the Michigan Medicine Translational Tissue Modeling Laboratory for providing enteroids and enteroid growth medium. The Translational Tissue Modeling Laboratory is a University of Michigan funded initiative (Center for Gastrointestinal Research, Office of the Dean, Comprehensive Cancer Center, Departments of Pathology, Pharmacology and Internal Medicine) with support by the Endowment for Basic Sciences. B.R. was supported by Penn State University (PSU)/National Institutes of Health (NIH) grant no. T32GM102057. S.L.C. and E.R.R. were supported by Rosalind E. Franklin Science Achievement Graduate Fellowships from the Eberly College of Science at PSU. Y.M.S. was supported by NIH grant nos. R01CA148828, R01CA245546 and R01DK095201, the University of Michigan Comprehensive Cancer Center Core (grant no. P30CA046592), the GI SPORE Molecular Pathology and Bio Sample Core (grant no. P50CA130810) and the Center for Gastrointestinal Research (grant no. DK034933). P.C.D. was supported by the Collaborative Microbial Metabolite Center (grant no. 1U24DK133658-01). B.S.Z. and G.D.W. are supported by NIH grant no. R01DK107565. The IGRAM study was also supported by an unrestricted donation from the American Beverage Foundation for a Healthy America to the Children's Hospital of Philadelphia to support the Healthy Weight Program and the NIH National Center for Research Resources Clinical and Translational Science Program (grant no. UL1TR001878). G.D.W. is supported by services provided by the H-MARC Core for the Center for Molecular Studies in Digestive and

Liver Diseases (P30 P30DK050306) as well as the Penn Center for Nutritional Science and Medicine. C.E.T., K.B. and G.D.W. are supported by the PennCHOP Microbiome Program. P.C.D. was further supported by grant no. R01DK136117-01. P.C.D. and S.S. were supported by Crohn's and Colitis Foundation awards (grant nos. 675191 and 623914) and the American Heart Association postdoctoral fellowship (19 POST34380588). E.R.R. and M.S.K. were supported by PSU/National Institute of Diabetes and Digestive and Kidney Diseases (grant no. T32DK120509). J.P.Z. was supported by NIH grant no. R35GM138369). J.E.B. was supported by NIH grant nos. R00AI147165 and R35GM151045. A.D.P. was supported by NIH grant nos. U01DK119702, S10OD021750 and R35ES035027 and the Pennsylvania Department of Health using Tobacco CURE funds. A.D.P. and J.P.V.H. were supported by United States Department of Agriigulture National Institute of Food and federal appropriations under project no. PEN04702. T.Y., K.W.K. and F.J.G. are supported by the National Cancer Institute Intramural Research Program. N.J.H. was supported by PSU. E.E.W. was supported by NIH grant no. R01GM125842 and PSU.

**Author contributions** B.R., S.L.C. and A.D.P. conceived the project. P.C.D. and E.C.G. provided standards, suggestions and feedback for the duration of the project. M.A.G. and J.P.V.H. performed luciferase reporter assays and edited the manuscript. E.R.R. and J.P.C. generated *bsh* mutants. S.S. and Y.M.S. performed organoid experiments. N.J.H. and E.E.W. helped with BSH purification and provided suggestions and feedback for the duration of the project. T.Y., K.W.K. and F.J.G. helped with bile acid analysis and interpretation. B.R., S.L.C., M.S.K. and J.E.B. performed gnotobiotic experiments. I.K., J.E.B., C.E.T., K.B., B.S.Z., G.D.W. and J.P.Z. provided data analysis and interpretation. D.D. and S.A. synthesized BBAA standards. B.R., S.L.C., N.J.H., E.E.W., P.C.D. and A.D.P. wrote the manuscript. All authors edited and approved the final manuscript.

**Competing interests** P.C.D. is an advisor to and holds equity in Cybele and is a scientific cofounder of, is an advisor to and holds equity in Ometa, Armome and Enveda, with previous approval by the University of California San Diego. J.V.H. is cofounder and chief scientific officer of INDIGO Biosciences, Inc., with previous approval from Penn State University. The other authors declare no competing interests.

## Additional information

**Correspondence and requests for materials** should be addressed to Andrew D. Patterson.

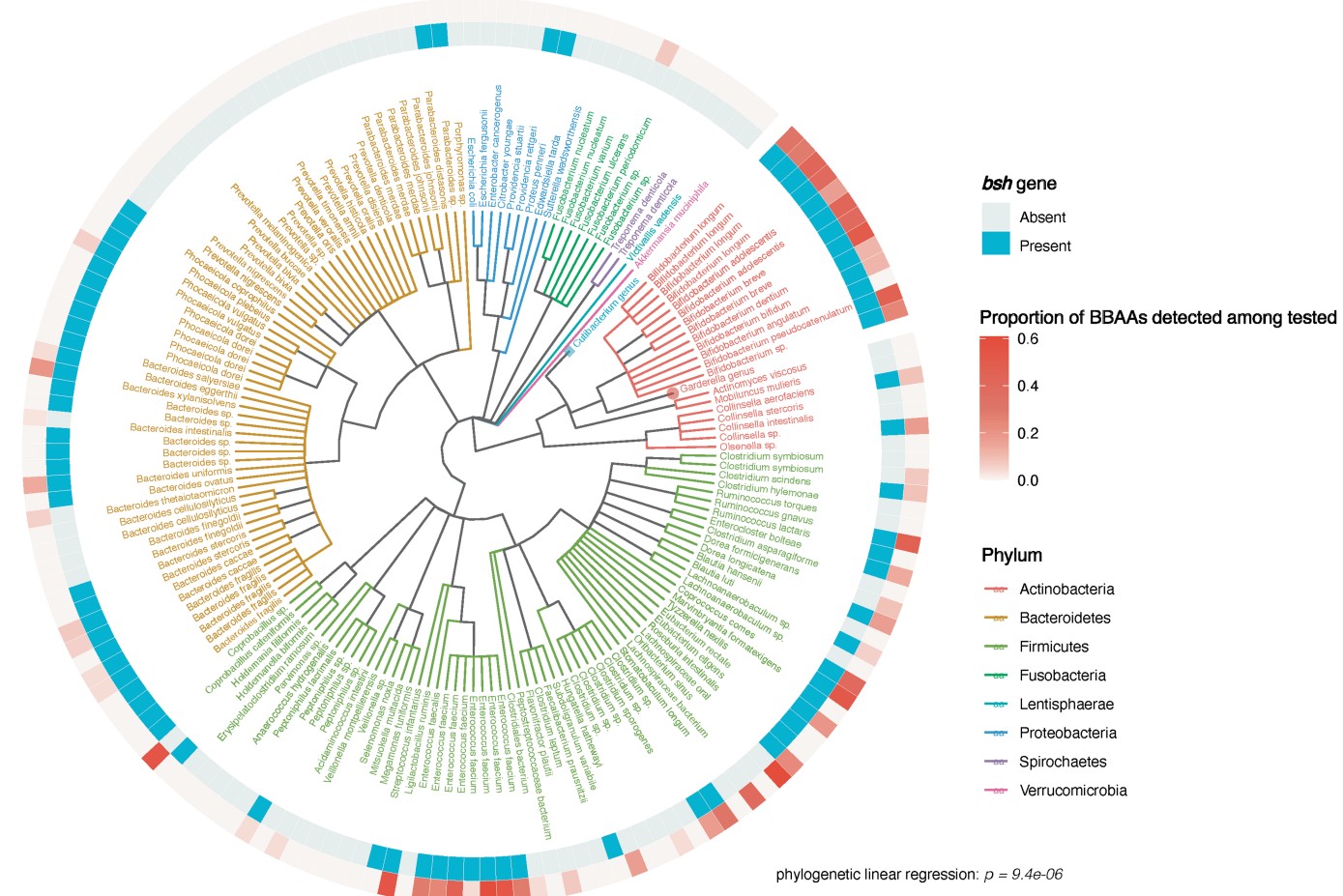

**Extended Data Fig. 1 | An extended version of correlation plot.** Plot of *bsh* presence in human-associated bacteria and BBAA production showing all the taxa names. Taxa-names are trimmed for readability. Two species of genera, *Cutibacterium* and *Gardnerella*, are condensed to their parent genus group.

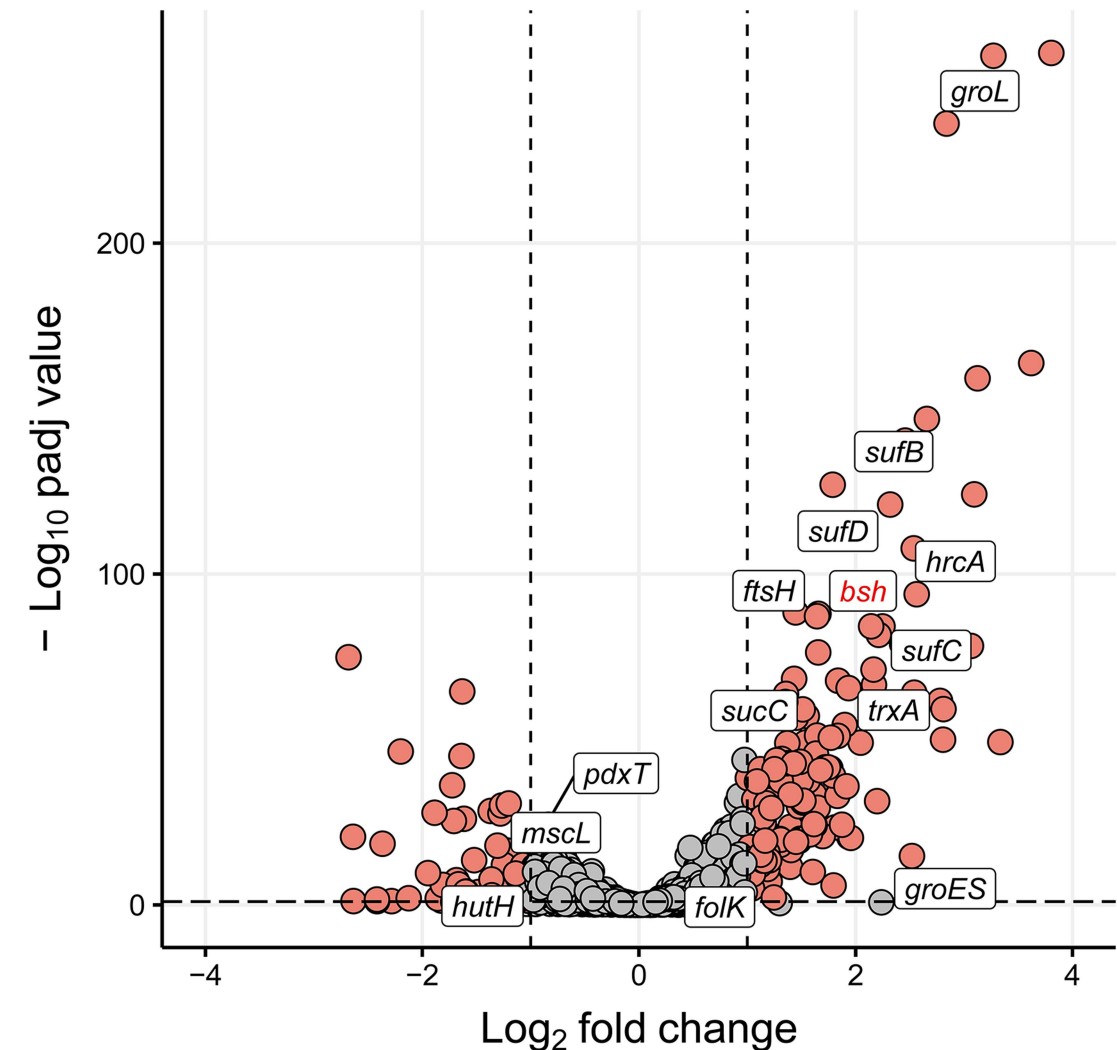

**Extended Data Fig. 2 | Differential gene expression analysis of *Bifidobacterium longum* NCTC 11818 supplemented with 200 μM each of CA and DCA.** Genes with absolute log$_2$ fold-change of 1 and FDR-adjusted $p \leq 0.05$ are represented in red (based on two-sided Wald tests as implemented in DESeq2); n = 5.

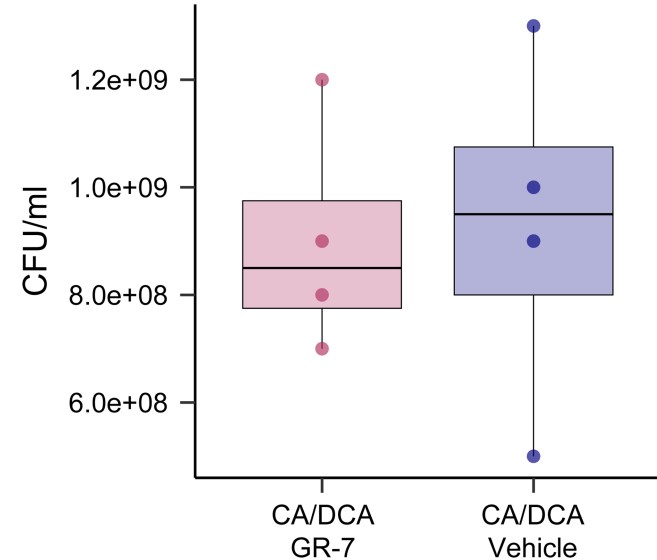

**Extended Data Fig. 3 | Colony-forming units of *Bifidobacterium longum* NCTC 11818 with or without GR-7.** The bounds of the box plot represent Q1–Q3 with median as center and min-max values for whiskers. n = 4 biologically independent bacterial cultures per group.

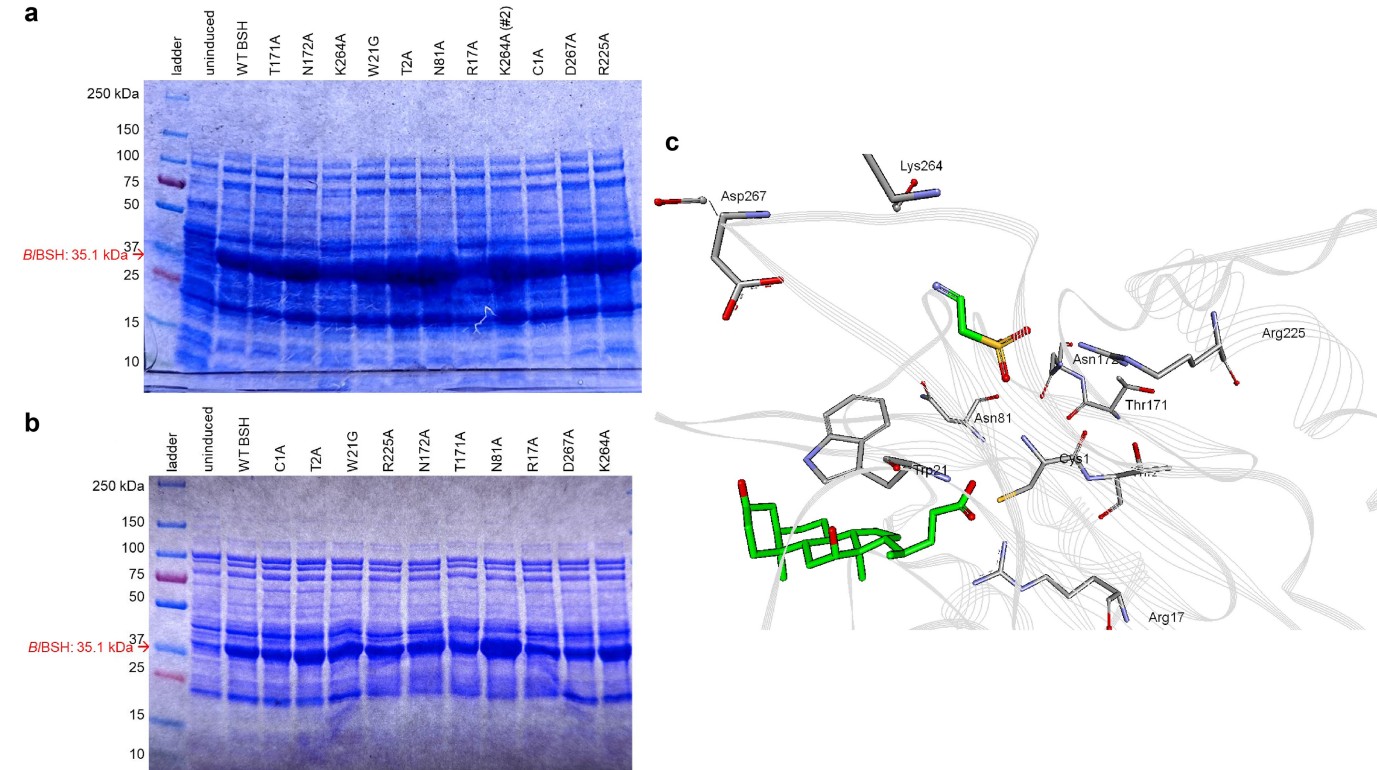

**Extended Data Fig. 4 | Heterologous protein expression of *Bl*BSH with active site residue substitutions.** SDS-PAGE from single experimental replicates showing *Bl*BSH expression at ~35 kDa, 4 h post-induction, in (**a**) LB media or (**b**) M9 minimal media. (**c**) Structure of *B. longum* BSH (PDB: 2HF0) highlighting active site residues mutated in this study in grey. Cholate and taurine (green) were fit using Discovery Studio software from the structure of *Clostridium perfringens* BSH (PDB: 2BJF) to demonstrate ligand binding location.

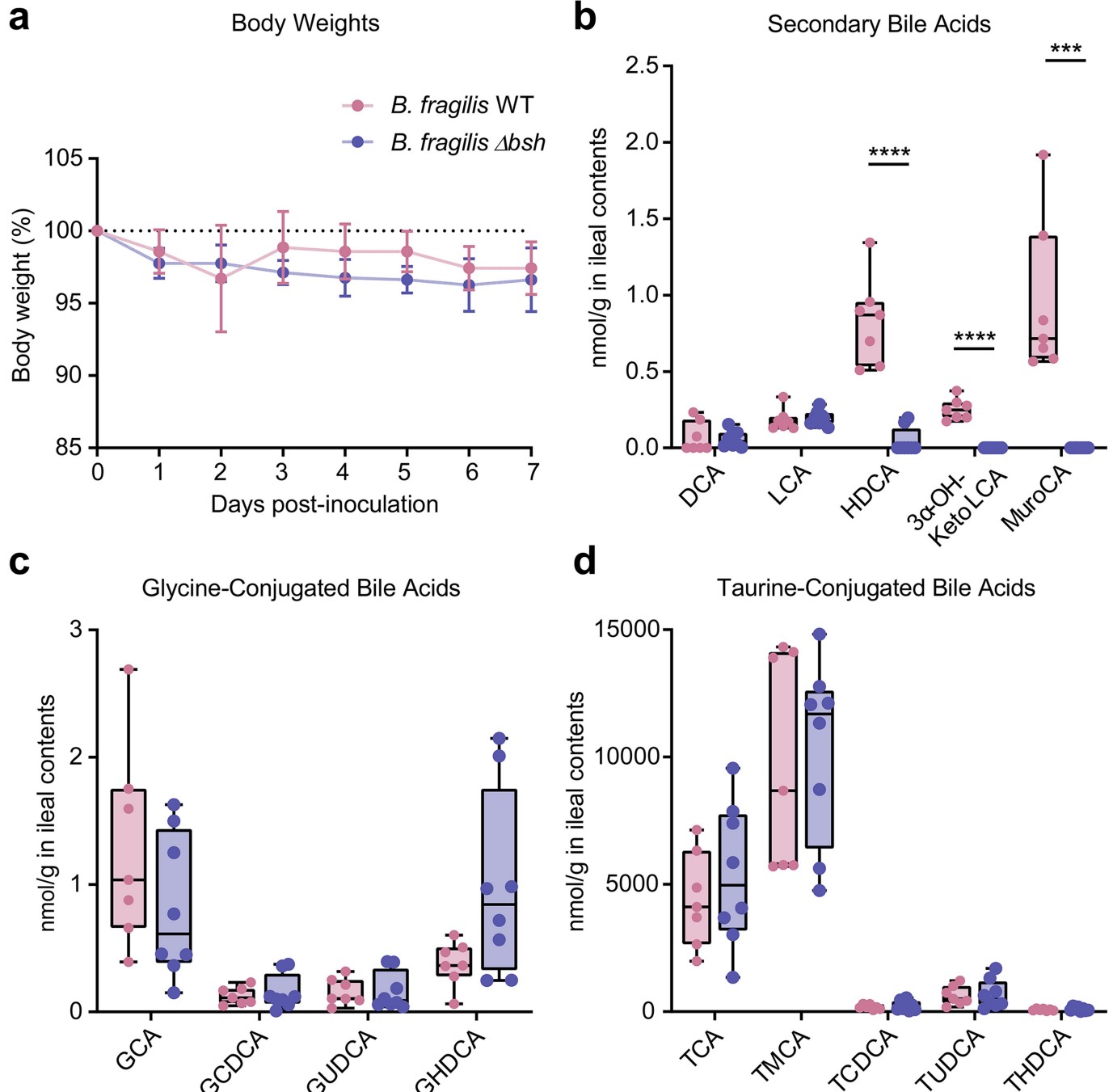

**Extended Data Fig. 5 | Body weights and ileal bile acid concentrations in *B. fragilis* monocolonized mice.** (**a**) Body weights relative to weight at inoculation on day 0. Data are presented as mean values ± SD. Ileal concentrations of (**b**) secondary, (**c**) glycine-conjugated, and (**d**) taurine-conjugated bile acids. Significant differences were determined by two-tailed t-test between WT (pink) and *Δbsh* (blue) colonized mice (HDCA $p$ = 5.2e-6****, 3αOH-KetoLCA $p$ = 1.1e-7****, MuroCA $p$ = 1.4e-4***). For (**b**-**d**), the bounds of the box plots represent Q1–Q3 with median as center and min-max values for whiskers. Biological replicates were n = 7 for WT and n = 8 for *Δbsh*.

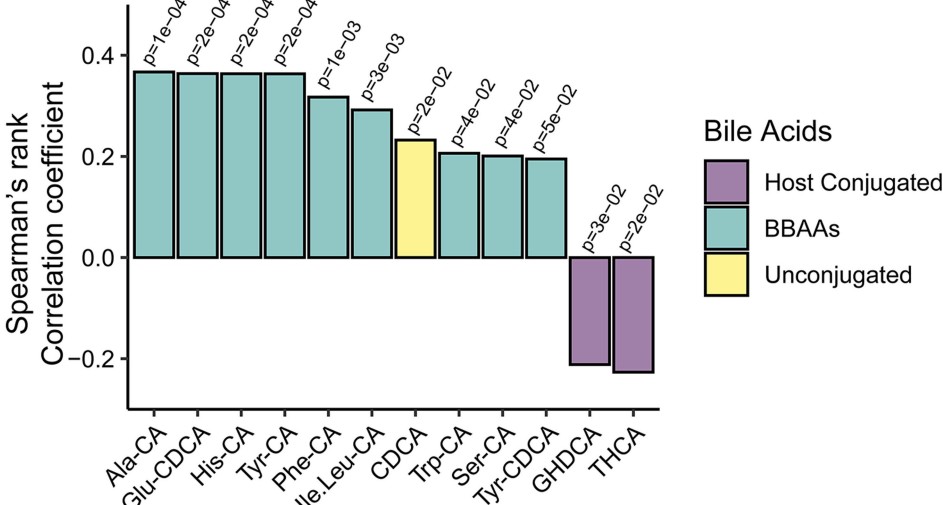

**Extended Data Fig. 6 | Levels of BBAAs correlate with the time elapsed after birth until sample collection within 0–4 day sampling group.** Spearman's rank correlation was used to calculate significant correlations between individual bile acids quantified by LC-MS/MS and hours after birth the initial sampling occurred.

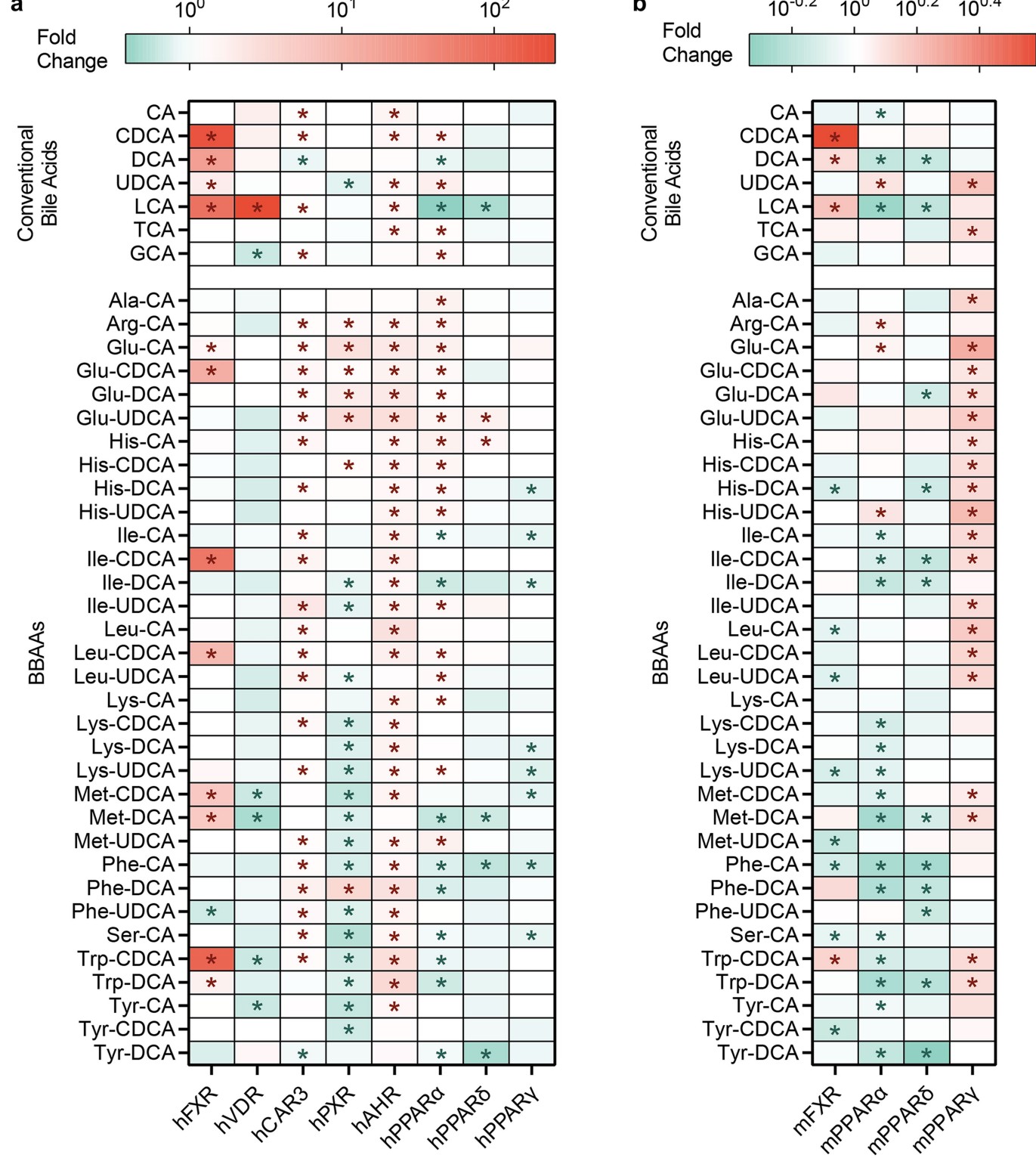

**Extended Data Fig. 7 | Regulation of human and murine ligand-activated transcription factors by bile acids.** (**a**) Luciferase reporter assays for activation of human transcription factors FXR, VDR, CAR3, PXR, AHR, and PPARα, β/δ, and γ by a 50 µM dose of conventional and bacterially conjugated bile acids. (**b**) Luciferase reporter assays for activation of mouse nuclear receptors FXR and PPARα, β/δ, and γ by a 50 µM dose of conventional and bacterially conjugated bile acids. Differences between treatments and vehicle-treated control were determined using ANOVA with Dunnett's post-hoc test ($p < 0.05$*), n = 3 per group. Color gradient is on a log scale.

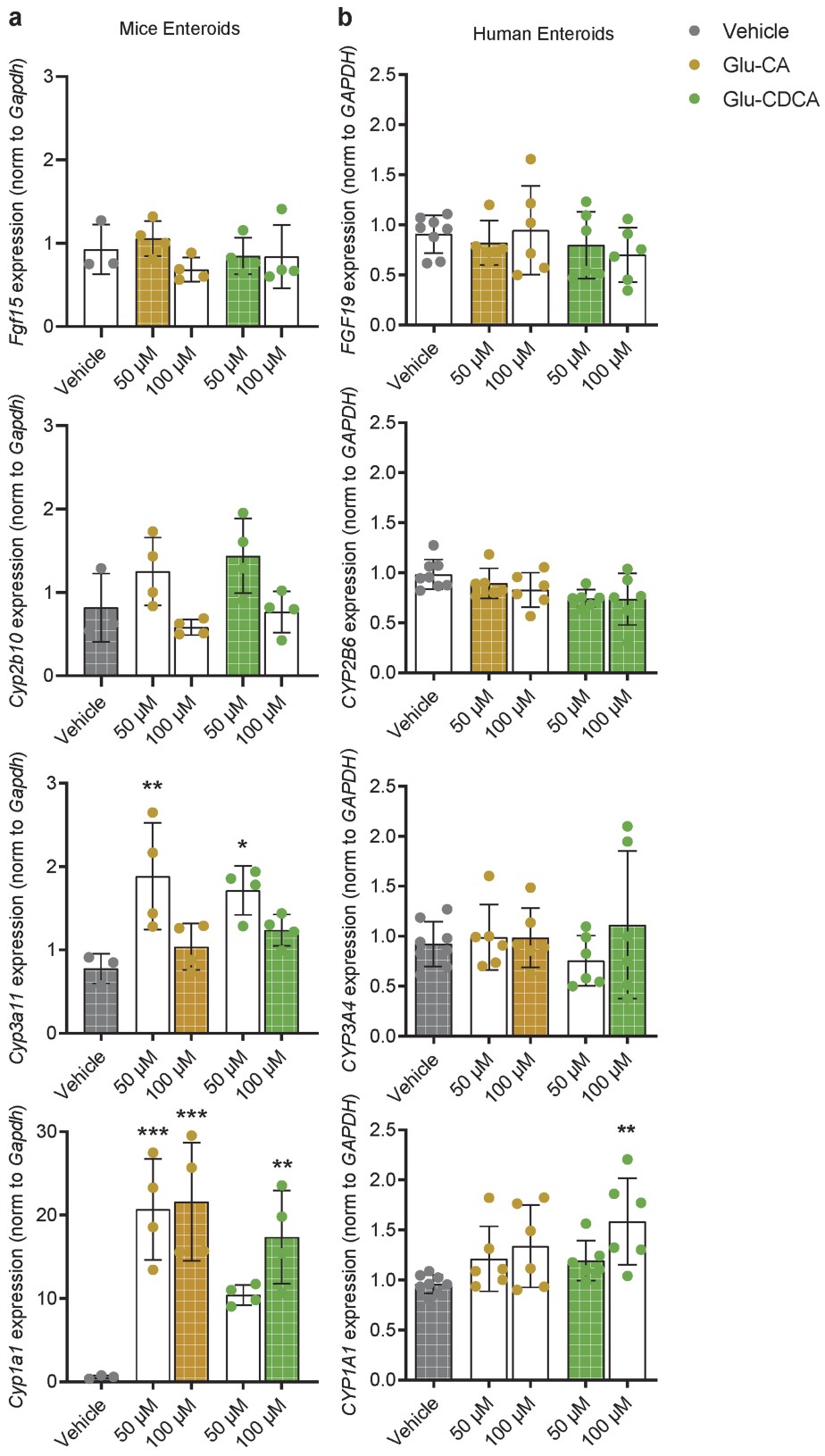

**Extended Data Fig. 8 | Regulation of ligand-activated transcription factors by BBAAs in organoids.** RT-qPCR of target genes for FXR (*Fgf15/FGF19*), CAR3 (*Cyp2b10/CYP2B6*), PXR (*Cyp3a11/CYP3A4*), and AHR (*Cyp1a1/CYP1A1*) in (**a**) mouse and (**b**) human small intestinal organoids treated with Glu-CA or Glu-CDCA. Expression was calculated using the $2^{-\Delta\Delta Ct}$ method, normalized to *Gapdh/GAPDH*. Error bars represent SD. (**a**) Biological replicates were n = 3 for vehicle and n = 4 for treatments (*Cyp3a11 p* = 4.8e-3**, *p* = 1.5e-2*, *Cyp1a1* Glu-CA-50 *p* = 4.7e-4***, Glu-CA-100 *p* = 3.3e-4***, Glu-CDCA-100 *p* = 2.3e-3). (**b**) Biological replicates were n = 8 for vehicle and n = 6 for treatments across two independent experiments (*p* = 3.3e-3**). All significant differences between treatments and vehicle were determined using one-way ANOVA with Dunnett's post-hoc test.

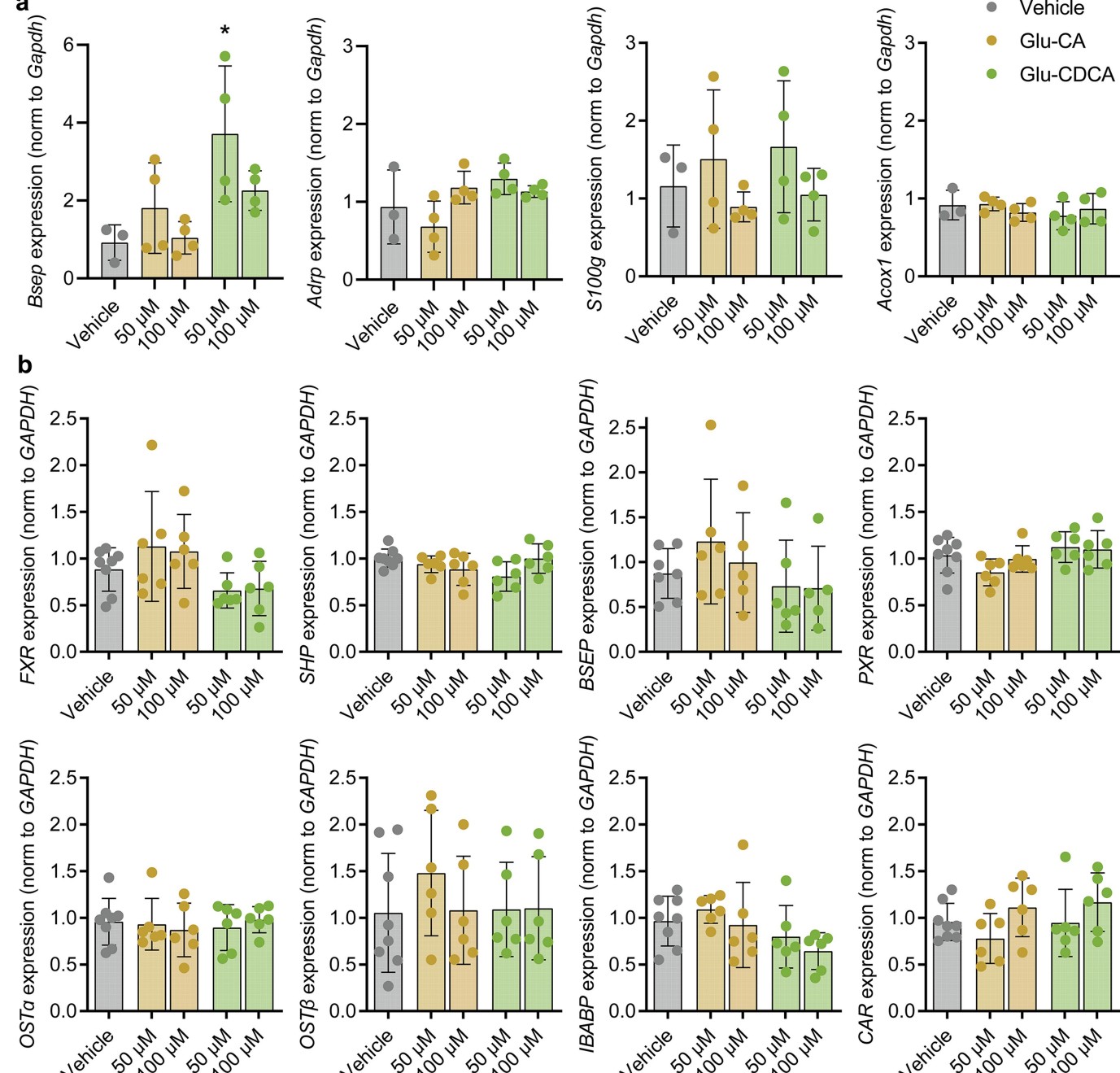

**Extended Data Fig. 9 | RT-qPCR of additional receptor target genes in organoids.** (**a**) Relative mRNA expression of *Bsep, Adrp, S100g,* and *Acox1* in mouse intestinal organoids. Biological replicates were n = 3 for vehicle and n = 4 for treatments. (**b**) Relative mRNA expression of *FXR, PXR, CAR,* and the FXR target genes short heterodimeric partner (*SHP*), bile salt export protein (*BSEP*), organic solute transporter-α (*OSTα*) and -β (*OSTβ*), and ileal bile acid binding protein (IBABP) in human intestinal organoids. Biological replicates were n = 8 for vehicle and n = 6 for treatments. Differences between treatments and vehicle were determined using one-way ANOVA with Dunnett's post-hoc test ($p = 0.011$*).

# Reporting Summary

## Statistics

For all statistical analyses, confirm that the following items are present in the figure legend, table legend, main text, or Methods section.

| n/a | Confirmed | |
|---|---|---|
| ☐ | ☒ | The exact sample size (*n*) for each experimental group/condition, given as a discrete number and unit of measurement |
| ☐ | ☒ | A statement on whether measurements were taken from distinct samples or whether the same sample was measured repeatedly |
| ☐ | ☒ | The statistical test(s) used AND whether they are one- or two-sided *Only common tests should be described solely by name; describe more complex techniques in the Methods section.* |
| ☐ | ☒ | A description of all covariates tested |
| ☐ | ☒ | A description of any assumptions or corrections, such as tests of normality and adjustment for multiple comparisons |
| ☐ | ☒ | A full description of the statistical parameters including central tendency (e.g. means) or other basic estimates (e.g. regression coefficient) AND variation (e.g. standard deviation) or associated estimates of uncertainty (e.g. confidence intervals) |
| ☐ | ☒ | For null hypothesis testing, the test statistic (e.g. *F*, *t*, *r*) with confidence intervals, effect sizes, degrees of freedom and *P* value noted *Give P values as exact values whenever suitable.* |
| ☒ | ☐ | For Bayesian analysis, information on the choice of priors and Markov chain Monte Carlo settings |
| ☐ | ☒ | For hierarchical and complex designs, identification of the appropriate level for tests and full reporting of outcomes |
| ☐ | ☒ | Estimates of effect sizes (e.g. Cohen's *d*, Pearson's *r*), indicating how they were calculated |

*Our web collection on statistics for biologists contains articles on many of the points above.*

## Software and code

Policy information about availability of computer code

| Data collection | Bruker TopSpin - ver. 4.0.9<br>Waters MassLynx - ver. 4.1<br>Thermo Xcalibur (TSQ Quantis Plus) - ver. 4.5.474.0<br>Thermo Xcalibur (Orbitrap Exploris 120) - ver. 4.6.67.17 |
|---|---|
| Data analysis | GraphPad Prism - ver. 6.07<br><br>Chenomx NMR Suite - ver. 9.02<br>Waters TargetLynx - ver. 4.1<br>Thermo Xcalibur (TSQ Quantis Plus) - ver. 4.5.474.0<br>Thermo Xcalibur (Orbitrap Exploris 120) - ver. 4.6.67.17<br>MS-DIAL- ver. 4.9.221218<br>R - ver. 4.2.2<br><br>The code used for analysis and generating the figures are found at GitHub (https://github.com/PattersonLab-PSU/BBAAs-paper) and Zenodo (10.5281/zenodo.10072966). |

For manuscripts utilizing custom algorithms or software that are central to the research but not yet described in published literature, software must be made available to editors and reviewers. We strongly encourage code deposition in a community repository (e.g. GitHub). See the Nature Portfolio guidelines for submitting code & software for further information.

## Data

Policy information about availability of data

All manuscripts must include a data availability statement. This statement should provide the following information, where applicable:

- Accession codes, unique identifiers, or web links for publicly available datasets
- A description of any restrictions on data availability
- For clinical datasets or third party data, please ensure that the statement adheres to our policy

RNA-sequencing datasets are deposited in NCBI SRA Database under bioproject: PRJNA878764. BBAA quantification data (Figure 1) were obtained from earlier published studies: doi.org/10.21203/rs.3.rs-820302/v1 and 10.1128/mSystems.00805-21 and corresponding bacterial genomes for comparative genomics were obtained from NCBI RefSeq (see Figure 1D source data and GitHub). For the human study, raw triple quadrupole mass spectrometry data have been deposited in the Mass Spectrometry Interactive Virtual Environment (MassIVE) under accession number MSV000093144. Structures of Bifidobacterium longum and Clostridium perfringens BSH were acquired from the Protein Database (PDB: 2HF0 and 2BJF, respectively). The Infant Growth and Microbiome (IGram) Study enrolled pregnant African American mothers and their newborn infants. The purpose of the research study was to learn more about the bacteria normally living in the child's gut, how it is transferred from mother to child, and whether it affects the child's growth in the first three years of life. The IGram data used in this publication were consistent with the stated purpose of the research. The IRB approved consent documents included language that allowed participants to indicate if they would like to have their information included in future research. Subjects may participate in the original research without their information (even if de-identified) being included in future research. Therefore, the data submitted to the repository (SRA accessions PRJNA1042647 and PRJNA557731) were limited to those individuals who consented to future use of their data and are not the entire data set used in the analyses presented here. To request the complete data set, contact Babette Zemel, PhD (zemel@chop.edu) or Kyle Bittinger, PhD (bittingerk@chop.edu), with a summary of how the data will be used and how it is consistent with the goals of the study. The request will be reviewed by the study team at the Children's Hospital of Philadelphia. If approved, the data will be made available within one month at no cost.

## Human research participants

Policy information about studies involving human research participants and Sex and Gender in Research.

| | |
|---|---|
| Reporting on sex and gender | 56 male and 52 female subjects were used for the human study. |
| Population characteristics | Pregnant African American women in their third trimester who had a pre-pregnancy BMI in the healthy or obese range and were enrolled, provided they were carrying singletons, and were free of medical conditions associated with glucose regulation, immunosuppressants, chronic inflammatory, or autoimmune diseases. Their infants were enrolled at birth if they were term, and free of chromosomal anomalies and conditions affecting growth and development. |
| Recruitment | Pregnant women receiving care in the obstetrics clinics at the Hospital of the University of Pennsylvania and who met the enrollment criteria were invited to participate. |
| Ethics oversight | The study protocol was reviewed and approved by the Committee for the Protection of Human Subjects (Internal Review Board) of the Children's Hospital of Philadelphia. |

Note that full information on the approval of the study protocol must also be provided in the manuscript.

# Field-specific reporting

Please select the one below that is the best fit for your research. If you are not sure, read the appropriate sections before making your selection.

☒ Life sciences          ☐ Behavioural & social sciences          ☐ Ecological, evolutionary & environmental sciences

For a reference copy of the document with all sections, see nature.com/documents/nr-reporting-summary-flat.pdf

# Life sciences study design

All studies must disclose on these points even when the disclosure is negative.

| | |
|---|---|
| Sample size | The metagenomics and bile acid analysis of the human study were secondary and exploratory so no power calculation was performed to determine sample size. The gnotobiotic experiment sample size was determined based on access to germ free mice due to housing and breading limitations. |
| Data exclusions | No data was excluded. |
| Replication | All experiments were successfully repeated at least two times. The human study was not repeated. |
| Randomization | Bile acid quantification samples were randomized prior to LC-MS/MS analysis. The human study was prospective so there was no subject randomization. Gnotobiotic mice were randomly assigned to the two treatment groups. |

| Blinding | Blinding was not part of the study protocol and was not used. |

# Reporting for specific materials, systems and methods

We require information from authors about some types of materials, experimental systems and methods used in many studies. Here, indicate whether each material, system or method listed is relevant to your study. If you are not sure if a list item applies to your research, read the appropriate section before selecting a response.

## Materials & experimental systems

| n/a | Involved in the study |
|---|---|
| ☒ | ☐ Antibodies |
| ☐ | ☒ Eukaryotic cell lines |
| ☒ | ☐ Palaeontology and archaeology |
| ☐ | ☒ Animals and other organisms |
| ☒ | ☐ Clinical data |
| ☒ | ☐ Dual use research of concern |

## Methods

| n/a | Involved in the study |
|---|---|
| ☒ | ☐ ChIP-seq |
| ☒ | ☐ Flow cytometry |
| ☒ | ☐ MRI-based neuroimaging |

## Eukaryotic cell lines

Policy information about cell lines and Sex and Gender in Research

| Cell line source(s) | Cell lines for the luciferase-based reporter assays were obtained from INDIGO Biosciences. |
|---|---|
| Authentication | Authentication was performed by INDIGO Biosciences. |
| Mycoplasma contamination | Cell lines are free of mycoplasma based on INDIGO Biosciences' report. |
| Commonly misidentified lines (See ICLAC register) | N/A |

## Animals and other research organisms

Policy information about studies involving animals; ARRIVE guidelines recommended for reporting animal research, and Sex and Gender in Research

| Laboratory animals | Male 8- to 10-week-old C57BL/6J mice were maintained under germ-free conditions in positive pressure isolators. |
|---|---|
| Wild animals | N/A |
| Reporting on sex | 15 male mice were used in the B. fragilis NCTC 9343 or the Δbsh mutant monocolonization experiments. |
| Field-collected samples | N/A |
| Ethics oversight | Gnotobiotic experiments were conducted under the Pennsylvania State University Institutional Animal Care and Use Committee approved protocol 202101826 |

Note that full information on the approval of the study protocol must also be provided in the manuscript.

