## [Peer Review File · Nature]

Manuscript Title: Bile Salt Hydrolase Catalyses Formation of Amine-Conjugated Bile Acids

Reviewer Comments & Author Rebuttals

Reviewer Reports on the Initial Version:

Referees' comments:

Referee #1 (Remarks to the Author):

A. Summary of the key results

The manuscript by Rimal et al describes bacterial bile salt hydrolase as the amino acid conjugating enzyme. Phylogenetic linear regression led to a significant association between taxa that generated BBAs and bsh. A few exceptions to BAA production and bsh were noted. RNA-Seq was performed on *B. longum* NCTC 11818 treated with unconjugated CA or DCA. Of the genes differentially expressed, bsh was among them. The pan BSH inhibitor GR-7 was able to significantly inhibit BBA formation at 100 μ M, but did not affect growth. The authors then moved to the genetically tractable *B. fragilis* NCTC 9343 strain to create a bsh knockout and complementation in trans strain. Deletion of bsh prevented BBA formation while complementation rescued BBA formation. Resting cell assays of *E. coli* ectopically expressing bsh conferred the phenotype of BBA formation. Purified BSH had a wider range of conjugation to amino acids than observed in whole cells, likely owing to intracellular concentrations of amino acids being limited. The mechanism of conjugation is left unexplored. The in vitro conjugation of amino acids by *B. fragilis* from 100 μ M TCA or CA is between 0.1 and 0.3 μ M. In *E. coli* expressing BSH, it is between 0.005 and 0.075 μ M. With pure enzyme it is between 0.03 to 0.06 μ M. To understand the physiological significance of BBAs, the authors screened 33 BBAs in a luciferase assay to measure agonism to FXR, VDR, CAR3, PXR, and AHR at a concentration of 50 μ M BBAs. Activation was dependent on the BA moiety as well as the amino acid conjugates. Organoids of human ileocytes were then screened with 50 or 100 μ M Glu-CA or Glu-CDCA. Glu-CA activated only AHR target gene CYP1A1 and only at 100 μ M. FXR was not activated in organoids. Overall, the combination of BSH inhibition of whole cells, gene knockout, and work with pure BSH enzyme convinces the reader that BSH is forming BBAs. The activity appears to be quite low; however, and additional quantitative enzyme data (kinetic constants) would be useful in understanding the extent of this process.

B. Originality and significance: if not novel, please include reference

C. Data & methodology: validity of approach, quality of data, quality of presentation

F. Suggested improvements: experiments, data for possible revision

BSH enzymes have been studied for decades, and given the current knowledge of the structure/mechanism of BSH, some attempt at determining the mechanism for conjugation would be expected. The Cys2 residue is an obvious example.

With the small amount of conjugating activity, as judged by Figure 3 in which 100 μ M TCA or CA addition yielded nanomolar levels of BBAs while host receptor activation required orders of magnitude higher levels leads to questions of physiological significance. Can the authors point to

data that indicates that 100 μ M BBAs is a physiologically relevant concentration?

Minor Point: The point of the RNA-Seq is unclear, since there doesn't seem to have been an attempt to look for potential candidate conjugating enzymes apart from BSH. It is well known that bsh is typically constitutively expressed in most bacterial species and strains.

In the enteroid model, the authors note a discrepancy in receptor agonism with the luciferase assay. It is known that human carboxypeptidases from pancreas cleave non-canonical amino acid-bile acid conjugates (PMID: 2867000). Enteroendocrine cells of the ileum express carboxypeptidases (PMID: 25051500). Did the authors check to see if the BBAs added to enteroids remained intact at the end of the experiment?

D. Appropriate use of statistics and treatment of uncertainties

E. Conclusions: robustness, validity, reliability

Unless the authors can physiologically justify the use of BBAs at 100 μ M, the conclusion that "BBAs may unexpectedly impact the metabolism of substrates, drugs, and carcinogens regulated by PXR and AHR" (line 224-225) is a stretch. Similar point at line 240 that "BBAs facilitate communication between the microbiota and host..."

G. Clarity and context: lucidity of abstract/summary, appropriateness of abstract, introduction and conclusions

The manuscript and figures are clearly presented.

Referee #2 (Remarks to the Author):

This manuscript the authors establish bacterial bile salt hydrolase as a bile acid amidating enzyme (an amine N-acyltransferase with bile acid as carboxylate co-substrate). This is an interesting manuscript: the experiments are thorough and the combination of biochemistry and genetic work make a compelling case that this enzyme is responsible for making BBAs in a variety of bacterial taxa.

The organoid assay is nice and its points of disagreement with the luciferase-based NHR activation assay are a cautionary note on the latter. My only point of concern is that I don't think these experiments go far enough in establishing the physiologic relevance of the BBAs. Not required, but I would strongly suggest that the authors test, e.g., wild-type vs mutant *B. fragilis* in a germ-free mouse (mono-colonized or in the context of a small defined community) to show an effect on the host - almost any reasonable/convincing effect would do.

Referee #3 (Remarks to the Author):

In this paper, Rimal and colleagues aimed at exploring the metabolism of bile acid by intestinal bacteria. They observed that bile salt hydrolase (BSH) exhibits an amine N-acyl transferase activity that conjugates amines to form bile acid amidates (BBAAs). Using different methods the authors demonstrated that BSH is required for BBAA production and that BBAAs is able to activate several host ligand-activated transcription factors including FXR, PXR, androstane receptor, and AhR.

The paper is clear and well written. I do not see any methodological issue.

The in vitro demonstration is convincing but human data and effects in vivo and in pathological models would reinforce the paper.

Author Rebuttals to Initial Comments:

We wish to thank the reviewers for their constructive feedback which has significantly improved the manuscript. We have made substantial revisions and added convincing new data from germ free mice and a human infant cohort that supports our conclusions and importantly broadens the appeal of this new mechanism for bile salt hydrolase.

Referee #1 (Remarks to the Author):

A. Summary of the key results

The manuscript by Rimal et al describes bacterial bile salt hydrolase as the amino acid conjugating enzyme. Phylogenic linear regression led to a significant association between taxa that generated BBAs and bsh. A few exceptions to BAA production and bsh were noted. RNA-Seq was performed on *B. longum* NCTC 11818 treated with unconjugated CA or DCA. Of the genes differentially expressed, bsh was among them. The pan BSH inhibitor GR-7 was able to significantly inhibit BBA formation at 100 μ M, but did not affect growth. The authors then moved to the genetically tractable *B. fragilis* NCTC 9343 strain to create a bsh knockout and complementation in trans strain. Deletion of bsh prevented BBA formation while complementation rescued BBA formation. Resting cell assays of *E. coli* ectopically expressing bsh conferred the phenotype of BBA formation. Purified BSH had a wider range of conjugation to amino acids than observed in whole cells, likely owing to intracellular concentrations of amino acids being limited. The mechanism of conjugation is left unexplored. The in vitro conjugation of amino acids by *B. fragilis* from 100 μ M TCA or CA is between 0.1 and 0.3 μ M. In *E. coli* expressing BSH, it is between 0.005 and 0.075 μ M. With pure enzyme it is between 0.03 to 0.06 μ M. To understand the physiological significance of BBAs, the authors screened 33 BBAs in a luciferase assay to measure agonism to FXR, VDR, CAR3, PXR, and AHR at a concentration of 50 μ M BBAs. Activation was dependent on the BA moiety as well as the amino acid conjugates. Organoids of human ileocytes were then screened with 50 or 100 μ M Glu-CA or Glu-CDCA. Glu-CA activated only AHR target gene CYP1A1 and only at 100 μ M. FXR was not activated in organoids. Overall, the combination of BSH inhibition of whole cells, gene knockout, and work with pure BSH enzyme convinces the reader that BSH is forming BBAs. The activity appears to be quite low; however, and additional quantitative enzyme data (kinetic constants) would be useful in understanding the extent of this process.

Query: BSH enzymes have been studied for decades, and given the current knowledge of the structure/mechanism of BSH, some attempt at determining the mechanism for conjugation would be expected. The Cys2 residue is an obvious example.

Response: *We performed site-directed mutagenesis of 9 key amino acids found within the binding pocket of the bile salt hydrolase enzyme to investigate the mechanism (see Figure 2e). We found that the same active site (Cys2 referred to herein as Cys1 following auto-catalytic cleavage) and similar residues previously reported (Arg17, Asn172, Arg225) are involved in the N-acyl transfer and hydrolysis reactions.*

Query: With the small amount of conjugating activity, as judged by Figure 3 in which 100 μM TCA or CA addition yielded nanomolar levels of BBAA while host receptor activation required orders of magnitude higher levels leads to questions of physiological significance. Can the authors point to data that indicates that 100 μM BBAA is a physiologically relevant concentration?

Response: *We agree with the reviewer that doses used in our studies are higher than what would be expected in vivo for an individual BBAA. However, we noted that the total concentration of BBAA can reach μM levels in human stool samples as found in our study and those recently published (Gentry et al. 2021; Aatsinki et al. 2023). Specifically, extrapolating concentrations from the in vivo small intestine sampling study (i.e., Capscan (Shalon et al. 2023)), we noted that BBAA in some individuals may reach nearly 85 μM . We wish to indicate that our goal here was to suggest that individual BBAA may activate host receptors although additional investigation is required to better understand how the local concentrations of BBAA mixtures impact host receptor activity.*

Aatsinki, Anna-Katariina, Santosh Lamichhane, Heidi Isokääntä, Partho Sen, Matilda Kråkström, Marina Amaral Alves, Anniina Kesitalo, et al. 2023. "Dynamics of Gut Metabolome and Microbiome Maturation during Early Life." <https://doi.org/10.1101/2023.05.29.23290441>.
Gentry, Emily, Stephanie Collins, Morgan Panitchpakdi, Pedro Belda-Ferre, Allison Stewart, Mingxun Wang, Alan Jarmusch, et al. 2021. "A Synthesis-Based Reverse Metabolomics Approach for the Discovery of Chemical Structures from Humans and Animals." <https://doi.org/10.21203/rs.3.rs-820302/v1>.
Shalon, Dari, Rebecca Neal Culver, Jessica A. Grembi, Jacob Folz, Peter V. Treit, Handuo Shi, Florian A. Rosenberger, et al. 2023. "Profiling the Human Intestinal Environment under Physiological Conditions." *Nature* 617 (7961): 581–91.

Query: The point of the RNA-Seq is unclear, since there doesn't seem to have been an attempt to look for potential candidate conjugating enzymes apart from BSH. It is well known that bsh is typically constitutively expressed in most bacterial species and strains.

Response: *The purpose of the RNA-seq experiment was to assess changes in gene expression of any genes (including bsh) following treatment with unconjugated bile acids. We identified a large number of genes differentially expressed to pursue all of them individually. In hindsight, we agree it seems superfluous to the story. To keep the flow of the story, RNA-seq results have been moved to the extended data.*

Query: In the enteroid model, the authors note a discrepancy in receptor agonism with the luciferase assay. It is known that human carboxypeptidases from pancreas cleave non-canonical amino acid-bile acid conjugates (PMID: 2867000). Enteroendocrine cells of the ileum express carboxypeptidases (PMID: 25051500). Did the authors check to see if the BBAA added to enteroids remained intact at the end of the experiment?

Response: *Indeed, the ileum can express carboxypeptidase E and the ileum enteroids used*

here also express carboxypeptidase E (data not shown). We appreciate the reviewer's concern that in some cases, BBAAAs could be de-amidated by active carboxypeptidase E; however, quantitatively measuring this activity in detail would entail overcoming major technical hurdles working with enteroids including access to small amounts of material, challenges dissociating metabolites from the matrigel substrate, as well as how the culturing conditions may influence metabolism of BBAAAs. There are also potential differences between the reporter lines and the enteroids including uptake/transport, metabolism, and receptor levels. Our goal by including this data was to demonstrate that BBAAAs could activate host receptors. Since our main focus was on establishing a new enzymatic activity for BSH, we now include the reporter and enteroid data as extended data. Further, we include the following in the manuscript: "However, the impact of other host and/or bacterial enzymes on BBAA metabolism and associated activity warrants further investigation."

Referee #2 (Remarks to the Author):

This manuscript the authors establish bacterial bile salt hydrolase as a bile acid amidating enzyme (an amine N-acyltransferase with bile acid as carboxylate co-substrate). This is an interesting manuscript: the experiments are thorough and the combination of biochemistry and genetic work make a compelling case that this enzyme is responsible for making BBAAAs in a variety of bacterial taxa.

The organoid assay is nice and its points of disagreement with the luciferase-based NHR activation assay are a cautionary note on the latter. My only point of concern is that I don't think these experiments go far enough in establishing the physiologic relevance of the BBAAAs. Not required, but I would strongly suggest that the authors test, e.g., wild-type vs mutant *B. fragilis* in a germ-free mouse (mono-colonized or in the context of a small defined community) to show an effect on the host - almost any reasonable/convincing effect would do.

Response: *We thank the reviewer for this helpful advice. We have addressed these concerns in two ways. First, as suggested, we conducted the germ-free experiment (see Figure 2f-j) where we have colonized germ-free recipients with either wildtype or Δ bsh *B. fragilis*. We noted that in mice colonized with Δ bsh *B. fragilis* there was a significant reduction in total BBAAAs as well as significantly reduced levels of primary bile acids and free taurine. Additionally, to help provide more human physiological relevance, we examined the levels of BBAAAs in human newborns and tracked these levels through the first year of life. We found that BBAAAs are found in newborn stool samples early in life and that these levels are consistent with colonization of the newborn gut with bsh-expressing bacteria. These new data strongly support that BSH is responsible for BBAA production.*

Referee #3 (Remarks to the Author):

In this paper, Rimal and colleagues aimed at exploring the metabolism of bile acid by intestinal bacteria. They observed that bile salt hydrolase (BSH) exhibits an amine N-acyl transferase activity that conjugates amines to form bile acid amidates (BBAAAs). Using different methods the

authors demonstrated that BSH is required for BBAA production and that BBAA is able to activate several host ligand-activated transcription factors including FXR, PXR, androstane receptor, and AhR.

The paper is clear and well written. I do not see any methodological issue.

The in vitro demonstration is convincing but human data and effects in vivo and in pathological models would reinforce the paper.

Response: *We thank the reviewer for this helpful advice. Please see our response to reviewer 2 above addressing the addition of human data.*

Reviewer Reports on the First Revision:

Referees' comments:

Referee #1 (Remarks to the Author):

The authors have addressed the major concerns of this reviewer, mainly with respect to demonstrating that the BSH enzyme is mechanistically responsible for conjugation. The functional data and in vivo quantification provide for biological relevance for BBAs. I have no further comments.

Referee #2 (Remarks to the Author):

The authors have done a reasonable job addressing my concerns and those of the other referees.

Referee #3 (Remarks to the Author):

The authors addressed my comments